# A Generalized Information Bottleneck Theory of Deep Learning

## Abstract

The Information Bottleneck (IB) principle offers a compelling theoretical framework to understand how neural networks (NNs) learn. However, its practical utility has been constrained by unresolved theoretical ambiguities and significant challenges in accurate estimation. In this paper, we present a *Generalized Information Bottleneck (GIB)* framework that reformulates the original IB principle through the lens of synergy, i.e., the information obtainable only through joint processing of features. We provide theoretical and empirical evidence demonstrating that synergistic functions achieve superior generalization compared to their non-synergistic counterparts. Building on these foundations we re-formulate the IB using a computable definition of synergy based on the average interaction information (II) of each feature with those remaining. We demonstrate that the original IB objective is upper bounded by our GIB in the case of perfect estimation, ensuring compatibility with existing IB theory while addressing its limitations. Our experimental results demonstrate that GIB consistently exhibits compression phases across a wide range of architectures (including those with *ReLU* activations where the standard IB fails), while yielding interpretable dynamics in both CNNs and Transformers and aligning more closely with our understanding of adversarial robustness.

## 1 Introduction

Deep learning has achieved remarkable practical success, yet our theoretical understanding of how neural networks learn effective representations remains incomplete (Shwartz-Ziv & Tishby, 2017). Information theory offers a principled framework for analyzing deep learning, as information-theoretic quantities are invariant to invertible transformations and provide interpretable units of measurement (Cover & Thomas, 1991). The Information Bottleneck (IB) principle, introduced by Tishby et al. (1999), has emerged as a particularly influential framework for understanding neural network learning dynamics, providing insights into diverse phenomena including adversarial robustness (Ma et al., 2021), the effects of dropout regularization (Achille & Soatto, 2018), and generalization bounds (Kawaguchi et al., 2023). Through this interpretation, the activations of our network can be viewed as a hidden state representation $\mathcal{T}$, which converges to a set of statistics captured by two competing terms. The first term, referred to as the *prediction term*, quantifies the mutual information (MI) between the hidden representation $\mathcal{T}$ and the target $Y$, denoted as $I(Y; \mathcal{T})$[1]. It is straightforward to see that achieving training (and consequently test) accuracy above random

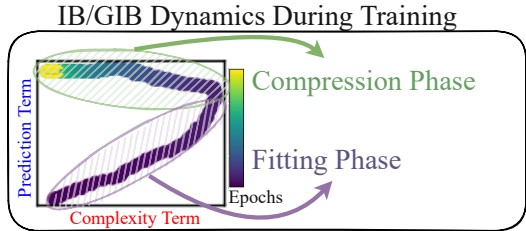

Figure 1: This schematic illustrates information plane dynamics during training, with trajectories color-coded from early epochs (light colors) to late epochs (dark purple), showing distinct fitting and compression phases.

---

[1]For the mathematical notation used throughout this paper, see Appendix A.

guessing requires a network whose learned representation is well aligned with that of the target data. However, it is well established that optimizing solely for prediction accuracy can lead to overfitting. Consequently, the IB framework introduces a second term: the *complexity term* $I(\mathcal{X}; \mathcal{T})$, which quantifies the mutual information between the input data $\mathcal{X}$ and the hidden representation. Optimizing this less intuitive term can be interpreted as an effort to minimize redundant and irrelevant information from the input that is encoded in the latent space.

Overall, the IB framework posits that deep neural networks learn by solving the following Lagrangian optimization problem:

$$\mathcal{L}_{\text{IB}} = \max_{p(\mathcal{T}|\mathcal{X})} \left[ \underbrace{I(\mathcal{T}; Y)}_{\text{prediction term}} - \underbrace{\beta^{-1} I(\mathcal{X}; \mathcal{T})}_{\text{complexity term}} \right] \tag{1}$$

Shwartz-Ziv & Tishby (2017) suggested that the success of deep learning can be attributed to the ability of NNs to perform the aforementioned optimization problem in two distinct phases. First, a *fitting phase*, where both of the two introduced terms increase, and second a *compression phase*, where the complexity term decreases in size (refer to Figure 1 for a visual illustration of these processes). It was argued that this second compression phase was unique to deep models and helped explain their generalizability. In some cases, the flow of information through the latent space has been shown to align precisely with the IB's theoretical predictions (Shwartz-Ziv & Tishby, 2017).

While the IB framework initially seemed to provide a complete explanation of how neural networks balance compression and predictive accuracy, Saxe et al. (2019) presented counterexamples that challenge this view. In particular, the authors showed that compression phases depend critically on the choice of activation function: networks with *tanh* activations exhibited compression across all layers, whereas *ReLU*-based networks did not. Despite the absence of a compression phase, the *ReLU* networks still generalized well. According to Goldfeld et al. (2019), this occurs because the complexity term in deterministic networks is theoretically constant or infinite, rendering compression impossible. Consequently, the compression observed in *tanh* networks was not a genuine information-theoretic effect, but rather the result of injected randomness (Saxe et al., 2019; Shwartz-Ziv & Tishby, 2017; Geiger, 2022).

In this paper, we address these issues by introducing a generalized formulation of the IB framework that is grounded in synergy. Synergy, a concept from multivariate information theory, captures the extra predictive power that arises when inputs are considered together rather than in isolation (Williams & Beer, 2010). To motivate this perspective, we begin by asking: why synergy? We then present both theoretical arguments and empirical results showing that synergistic functions lead to improved generalization.

Having established that synergistic functions generalize better, we construct the GIB by reformulating the IB through the lens of synergy. First, we introduce a point-wise mutual information (PMI)-based reweighting scheme that ensures we measure synergy specifically for correct predictions rather than arbitrary outputs. We then combine this reweighting with our feature-wise synergy decomposition, which uses the interaction information (II) to quantify information available only through joint processing of all features. Finally, we cast this as a Lagrangian optimization problem, yielding our GIB objective that measures how synergistically the input features combine to describe correct outputs.

After deriving the GIB, we prove that, under perfect estimation, it can be lower bounded by the IB. Importantly, our formulation overcomes key theoretical limitations of standard IB, including the issue of infinite complexity terms. We demonstrate that GIB exhibits clear compression phases and interpretable learning dynamics across a wide range of scenarios where standard IB fails. In Figure 2, we revisit the experiments presented in Saxe et al. (2019) and show that the GIB displays compression phases for five different activation functions while the IB is limited to one[2]. Beyond these synthetic examples, we observe consistent

---

[2]**Reading Information Planes Plots.** Throughout, we visualize information dynamics using information plane plots. In these plots, the $x$-axis represents the complexity term and the $y$-axis

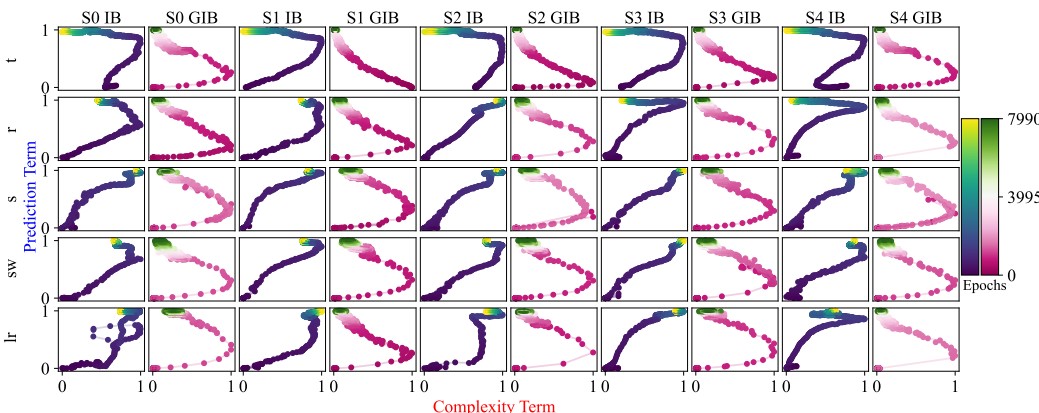

Figure 2: Information plane dynamics across multiple activation functions, extending Shwartz-Ziv & Tishby (2017) and Saxe et al. (2019) beyond *tanh* and *ReLU* to include *softplus*, *swish*, and *leaky ReLU*. Standard IB (blue) shows compression only for *tanh*; GIB (pink) shows compression for all activation functions. Each column represents one seed.

information dynamics in practical deep learning settings including ResNets on CIFAR-10 and BERT fine-tuning. Furthermore, the complexity term in our framework provides meaningful insights into model behavior under adversarial attacks, correctly tracking vulnerability where standard IB fails. Code for full reproducibility of these results will be made publicly available upon publication.

## 2 RELATED WORK

**MI Estimation.** MI estimation in the IB framework remains an active area of research and debate. Numerous estimators exist (e.g., $k$-nearest-neighbor (kNN) methods (Kozachenko & Leonenko, 1987; Kraskov et al., 2004), and kernel-density approaches (Kandasamy et al., 2015; Han et al., 2017)), and trainable neural estimators (Belghazi et al., 2018)), yet many information-theoretic studies of deep networks (Shwartz-Ziv & Tishby, 2017; Saxe et al., 2019) discretize neuron outputs ("binning") to approximate MI. Binning is simple and fast, but even moderate coarse-graining can introduce substantial estimation error (Goldfeld et al., 2019). Despite these limitations, we use binning because MI estimates are needed throughout training (e.g., at each epoch); running kNN, KDE, or variational estimators at this frequency would be prohibitively slow and numerically unstable in high-dimensional settings. Our goal is to track relative trends in MI rather than obtain exact differential values, and binning provides a tractable, reproducible proxy that makes per-training-step MI monitoring feasible.

**Other Generalizations of the Information Bottleneck.** There exist many extensions of the IB framework. For instance, the variational information bottleneck (VIB) introduces stochastic neural parameterizations to scale IB to deep networks (Alemi et al., 2017), while Information dropout applies a neuron-wise IB-like penalty via multiplicative noise to improve generalization (Achille & Soatto, 2018). The HSIC bottleneck (Wang et al., 2021) replaces mutual information with the Hilbert–Schmidt independence criterion to regularize intermediate representations for adversarial robustness. More recent work has extended the IB to multivariate and deep settings. Matrix-based Rényi's $\alpha$-order entropy functionals (Yu

---

represents the prediction term. For standard IB, these are $I(\mathcal{X};\mathcal{T})$ and $I(\mathcal{T};Y)$, respectively. For the full formulation of the GIB, see Section 3. Trajectories are color-coded by training epoch, progressing from early training (dark pink/blue) to late training (light green/yellow). Blue trajectories correspond to the standard IB dynamics, whereas pink trajectories depict the dynamics under our GIB formulation. Movement leftward indicates compression (reduction of redundant information), while movement upward indicates improved prediction. For the IB we only report the information plane of the final layer as this is where compression dynamics are most readily observed. Meanwhile, the GIB is formulated based on inputs and therefore only produces one information plane per training. For clarity of presentation, we normalize the complexity term and prediction term results between 0 and 1.

et al., 2020; 2021a) provide estimators of multivariate entropies and dependence. These have been used to develop deterministic IB objectives in deep networks (Yu et al., 2021b), and to design gated IB objectives for sequential environments (Alesiani et al., 2023). The multivariate IB of Friedman et al. (2001) captures statistical dependencies between multiple bottleneck variables through graphical models. Most relevant to our work, recent advances have also connected IB and Partial Information Decomposition (PID), with Kolchinsky (2024) building on earlier IB work such as Kolchinsky et al. (2019), showing that PID redundancy can be isolated via an IB-style optimization. Although these methods extend IB to multivariate settings and even show that certain PID quantities, such as redundancy, can be characterized via bottlenecks, none explicitly incorporate a synergy-specific information term into the IB formulation. Our approach achieves this, and we will demonstrate that it not only results in more consistent compression behavior but also offers theoretical advantages.

**Synergy.** Synergy characterizes the additional information obtained by evaluating variables collectively rather than individually, quantifying how features interact to reduce uncertainty about a target. The characteristics of this relationship can be illustrated by means of the XOR function. Consider two binary string variables, $X_1$ and $X_2$, with $Z$ being their XOR output. In this scenario, $X_1$ and $Z$, as well as $X_2$ and $Z$, are uncorrelated ($I(X_1; Z) = I(X_2; Z) = 0$), but together, $X_1$ and $X_2$ fully describe $Z$ ($I(X_1, X_2; Z) = H(Z)$) (Guyon & Elisseeff, 2003; Williams & Beer, 2010). While the concept is intuitive, its formalization has proven challenging, leading to multiple proposed measures. Early work by Williams & Beer (2010) introduced PID, which decomposes MI into unique, redundant, and synergistic components. However, the number of terms in this decomposition equals the $n − 1$'th Dedekind number, where $n$ is the number of features. This number is impractically large: a system with nine variables would require approximately $5 \times 10^{22}$ terms, while for ten variables, the Dedekind number remains unknown. Moreover, estimating these terms is subject to convergence issues and size limitations (Makkeh et al., 2018; 2019; Pakman et al., 2021). While Varley & Hoel (2022) reduced the number of investigable quantities by averaging contributions of layers in the PID lattice, with different layers representing different levels of redundancy or synergy, calculations remained too complex for applications involving more than a few features. Alternative measures such as O-information (Rosas et al., 2019), correlational importance (Nirenberg & Latham, 2003), and synergistic MI (Griffith & Koch, 2014) can estimate the synergy or redundancy of large sets of variables, but fail to reveal whether a specific feature interacts synergistically or redundantly. We resolve these issues by averaging the interaction information of each feature with those remaining:

$$\text{Syn}(\mathcal{X} \rightarrow Y) = I(\mathcal{X}; Y) - \frac{1}{N} \sum_{i=1}^{N} \left( I(\mathcal{X}^{-i}; Y) + I(X^i; Y) \right) \tag{2}$$

where $\mathcal{X}^{-i} = \mathcal{X} \backslash \{X^i\}$. This formulation captures how features collectively reduce uncertainty about the target $Y$, while maintaining computational feasibility by avoiding the exponential explosion of subset calculations (Westphal et al., 2025).

## 3 THE GENERALIZED INFORMATION BOTTLENECK

We now introduce the GIB, by first showing that, given two functions with identical mutual information (MI) with noisy training data, the function exhibiting higher synergy achieves tighter generalization bounds. This result motivates the principle that synergistic functions generalize better than non-synergistic ones, and thus learning should favor synergy. At the same time, we emphasize that functions must also be correct. To capture this, we introduce a distribution that prioritizes accurate predictions. The final formulation of the GIB therefore maximizes the synergistic contribution of the inputs in describing this distribution.

### 3.1 SYNERGY AND GENERALIZATION

In this subsection, we formally establish the connection between synergy and generalization. We begin by presenting theoretical results supported by experiments on synthetic data, and then extend the discussion to empirical findings on ResNets.

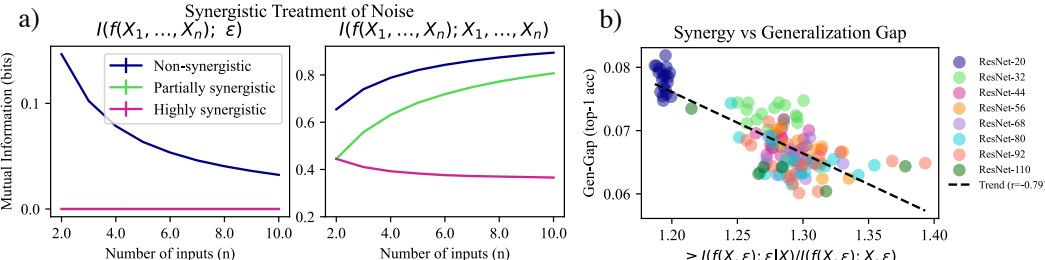

Figure 3: Synergistic processing of noise enhances generalization. (a) Controlled synthetic experiment demonstrating how synergy affects information flow (see Appendix E.1 for details). Three functions of increasing synergy process binary inputs with noise: non-synergistic (blue), partially synergistic (green), and highly synergistic (magenta). Left: $I(f(X,\varepsilon);\varepsilon)$: more synergistic functions have lower dependence on noise. Right: $I(f(X,\varepsilon);X)$ - we observe that synergistic functions have lower MI with the input. (b) Empirical validation on CIFAR-10 using ResNets of varying depths (see Appendix E.2). We quantify synergistic interactions between inputs and noise as $I(f(X,\varepsilon);\varepsilon|X)/I(f(X,\varepsilon);X,\varepsilon)$. Higher synergy correlates with smaller generalization gaps.

### 3.1.1 THEORETICAL EVIDENCE

Let us first suppose we have some noise $\varepsilon$ that can be considered independent of our input data $\mathcal{X}$. Now consider two functions $\mathrm{s}^-$ and $\mathrm{s}^+$. If $\mathrm{s}^+$ combines the independent components of its arguments in a more synergistic manner than $\mathrm{s}^-$, by definition we have:

$$I(\mathrm{s}^+(\mathcal{X},\varepsilon);\mathcal{X},\varepsilon) - I(\mathrm{s}^+(\mathcal{X},\varepsilon);\varepsilon) - I(\mathrm{s}^+(\mathcal{X},\varepsilon);\mathcal{X}) >$$
$$I(\mathrm{s}^-(\mathcal{X},\varepsilon);\mathcal{X},\varepsilon) - I(\mathrm{s}^-(\mathcal{X},\varepsilon);\varepsilon) - I(\mathrm{s}^-(\mathcal{X},\varepsilon);\mathcal{X}). \quad (3)$$

If we now assume that $I(\mathrm{s}^+(\mathcal{X},\varepsilon);\mathcal{X},\varepsilon) = I(\mathrm{s}^-(\mathcal{X},\varepsilon);\mathcal{X},\varepsilon)$ (which can crudely be thought of as approximately equal train accuracies) then it must be true that:

$$I(\mathrm{s}^+(\mathcal{X},\varepsilon);\varepsilon) + I(\mathrm{s}^+(\mathcal{X},\varepsilon);\mathcal{X}) < I(\mathrm{s}^-(\mathcal{X},\varepsilon);\varepsilon) + I(\mathrm{s}^-(\mathcal{X},\varepsilon);\mathcal{X}) \quad (4)$$

where $I(\mathrm{s}(\mathcal{X},\varepsilon);\varepsilon)$ represents the MI between the output of a function and the noise, while $I(\mathrm{s}(\mathcal{X},\varepsilon);\mathcal{X})$ describes the information shared between the output and uncorrupted input. In Figure 3(a) we analyze the implications of Equation 4 via synthetic data. We show that more synergistic functions for the same complexity of input and output have lower values of both $I(\mathrm{s}(\mathcal{X},\varepsilon);\varepsilon)$ and $I(\mathrm{s}(\mathcal{X},\varepsilon);\mathcal{X})$. This is favorable as both of these terms are known to be inversely related to generalization capabilities, as discussed below.

**How $I(\mathrm{s}(\mathcal{X},\varepsilon);\mathcal{X})$ Impedes Generalizability.** This quantity can be re-written as the complexity term of the IB, reducing its value has repeatedly been shown to be related to compression and generalization (Tishby et al., 1999; Shwartz-Ziv & Tishby, 2017). High values of this term ensure a latent representation that has memorized irrelevant and redundant information in the input. Furthermore, recent work has formally related this quantity to generalization bounds (Kawaguchi et al., 2023).

**How $I(\mathrm{s}(\mathcal{X},\varepsilon);\varepsilon)$ Impedes Generalizability.** The relationship between noise sensitivity and generalization is fundamentally tied to function smoothness. Most generalization bounds require that the learned function be *Lipschitz smooth*, meaning there exists a constant $L$ such that $\|f(x_1) - f(x_2)\| \le L\|x_1 - x_2\|$ for all inputs. This constraint ensures the function's output changes at most proportionally to input perturbations. When a function has high mutual information with noise $I(\mathrm{s}(\mathcal{X},\varepsilon);\varepsilon)$, it indicates the output varies significantly with small noise perturbations, implying a large Lipschitz constant. As shown by Bartlett et al. (2017) and Neyshabur et al. (2017), generalization bounds scale with the Lipschitz constant of neural networks, which can be bounded by the product of layer-wise spectral norms. Therefore, functions with lower $I(\mathrm{s}(\mathcal{X},\varepsilon);\varepsilon)$ exhibit smaller Lipschitz constants and tighter generalization bounds, explaining why synergistic functions that minimize noise sensitivity achieve superior generalization.

### 3.1.2 Empirical Evidence

To empirically validate our theoretical findings, we conducted experiments examining how synergistic processing of noise affects generalization in deep NNs. We trained ResNet models of varying depths (20, 32, 44, 56, 68, 80, 92, and 110 layers) on CIFAR-10 with standard data augmentations. To quantify synergy with augmentation noise, we developed a novel teacher-student framework: a teacher model trained with augmentations (random crops and horizontal flips) teaches a student model to predict its outputs from non-augmented inputs. The cross-entropy loss achieved by the student provides a maximal upper bound for the proportion of information between inputs and outputs that cannot be explained without considering the interaction of noise and features, formally: $I(f(X,\varepsilon);\varepsilon|X)/I(f(X,\varepsilon);X,\varepsilon)$.

Our results, shown in Figure 3(b), reveal a strong negative correlation (Pearson $r = -0.79$, $p < 0.001$) between this synergy measure and generalization performance across all model configurations. Models with higher synergy (those whose predictions depend more on the interaction between image content and augmentation patterns) consistently achieve smaller generalization gaps. This confirms our theoretical prediction: synergistic processing of augmentation noise, rather than treating it as independent corruption, enables models to extract more robust features that generalize better to clean test images. Full experimental details are provided in Appendix E.2. Considering that to synergistically process noise, we must synergistically process the features, we design our GIB principle based on measures of feature synergy.

### 3.2 Formulating the GIB Principle

During the last section, we argued that synergistic functions generalize better than their non-synergistic counterparts. Consequently, we argue that when learning, a deep network should aim to maximize the synergy of the inputs to produce the outputs. However, this is not a strict enough condition, because there are many different synergistic functions, most of which are irrelevant to the task at hand. We instead want to measure how synergistically our inputs combine to give the *correct* outputs.

To facilitate this, we take the following two steps. First, our prediction term will solely measure the MI between our predictions and targets. Second, our complexity term will be a function of a new distribution that describes the co-occurrences of $Z$ with $Y$. The exact definition of $Q$ is based on PMI-based reweighting, i.e., weighting samples by the likelihood ratio between the joint distribution and the product of marginals: $Q(Z,Y) = \frac{P(Z,Y)}{P(Z)P(Y)}$. This reweighting scheme emphasizes patterns where $Z$ and $Y$ co-occur more frequently than would be expected under independence, effectively highlighting the meaningful dependencies between our learned representations and the target outputs. PMI has proven effective in capturing meaningful associations in numerous ML contexts: it underlies word2vec's implicit matrix factorization (Levy & Goldberg, 2014), drives contrastive learning objectives (van den Oord et al., 2018), and measures feature relevance in interpretable ML (Bouma, 2009). By sampling from the distribution $Q(Z,Y) = \frac{P(Z,Y)}{P(Z)P(Y)}$ we obtain the random variable $Q$. Combining these steps with how we earlier defined synergy, we get the following formulation of the GIB:

$$\mathcal{L}_{\text{GIB}} = \max_{p(Z|X)} \left[ \underbrace{I(Z;Y)}_{\text{prediction term}} - \underbrace{\frac{1}{2\beta N} \sum_{i=1}^{N} \left( I(\mathcal{X}^{-i};Q) + I(X^i;Q) \right)}_{\text{complexity term}} \right] \tag{5}$$

The prediction term (blue) $I(Z;Y)$ measures the mutual information between the model outputs $Z$ and the labels $Y$, capturing how well the predictions align with the true targets. The complexity term (red) $\frac{1}{2\beta N} \sum_{i=1}^{N} \left( I(\mathcal{X}^{-i};Q) + I(X^i;Q) \right)$ inversely quantifies the average information obtainable from individual features or their complements about the PMI-reweighted distribution $Q(Z,Y)$, which emphasizes correct predictions. By maximizing their difference, GIB measures information dynamics that emerge only from collective feature

interactions, which our analysis also indicates leads to improved generalization. On the other hand, measuring synergy can be computationally demanding, as we discuss in Appendix G.

## 4 RELATING THE GIB TO THE IB

In this section, we first prove that under a simple assumption (i.e., perfect estimation) the IB is a lower bound of our GIB. Finally, we discuss how the GIB solves longstanding IB issues.

**Theorem 1.** *If we assume perfect training accuracy and therefore $Q = Z = Y$, then the original IB objective is upper bounded by our GIB:*

$$I(\mathcal{T}; Y) - \beta I(\mathcal{X}; \mathcal{T}) \leq I(Z; Y) - \frac{1}{2\beta N} \sum_{i=1}^{N} \left( I(\mathcal{X}^{-i}; Q)) + I(X^i; Q)) \right) \tag{6}$$

The proof is provided in Appendix B. This result demonstrates that the GIB provides an upper bound on the IB objective. Consequently, as we optimize the traditional IB to find sufficient statistics, we simultaneously optimize our GIB objective, ensuring that our approach remains compatible with the theoretical foundations of the IB. For instance, in Appendix C we prove the GIB discovers sufficient statistics.

This new formulation overcomes two main limitations of the original IB. First, the partition across subsets of features combined with the PMI definition of $Q(Z, Y)$ protects the compression term from becoming infinite. In Appendix D we prove that the GIB is only infinite under interpretable circumstances. Second, and more fundamentally, our formulation explicitly considers over-reliance on individual features, as explained in Section 3.1. IB optimizes the total information flow between inputs and outputs through the latent representation $\mathcal{T}$ without considering how features interact. In contrast, GIB explicitly models how inputs combine to form the latent representation, distinguishing between different types of feature interactions. This is evident in the complexity terms: IB's $\beta^{-1} I(\mathcal{X}; \mathcal{T})$ aggregates all information equally, while GIB's synergistic decomposition $\frac{1}{2\beta N} \sum_{i=1}^{N} \left( I(\mathcal{X}^{-i}; Q) + I(X^i; Q) \right)$ penalizes the information contained in individual features. Consequently, IB compresses indiscriminately, whereas GIB selectively preserves long-range feature combinations; the synergistic patterns we have shown lead to better generalization.

## 5 EXPERIMENTAL CASE STUDIES

This section presents experimental evidence demonstrating GIB's advantages over standard IB across diverse settings. We show that GIB provides more consistent and interpretable information dynamics throughout training, successfully capturing compression phases where standard IB fails (refer to Footnote 2 on reading information planes). Additionally, we demonstrate that GIB's complexity term serves as a direct indicator of adversarial vulnerability, providing quantitative insights into model robustness that standard IB cannot capture. As stated in Section 2, all MIs will be estimated using binning. However, in Appendix F, we reproduce our results with a different method of MI estimation.

### 5.1 INFORMATION DYNAMICS OF MLPS LEARNING SIMPLE FUNCTIONS

We examine NNs learning five mathematical functions of increasing complexity: addition, multiplication, and three symmetric polynomials labelled f1, f2 and f3 (polynomials in which all arguments are subjected to the same operations). Full experimental details are in Appendix E.3.

The information plane dynamics in Figure 4 show clear differences between the standard IB and GIB formulations. For the GIB, we observe compression phases, characterized by leftward movement during training, across all five functions and random seeds. The trajectories initially move upward and rightward as networks fit the training data, then shift leftward as training progresses. The standard IB displays more variable behavior, without real indication of compression, despite the strong generalization capabilities of these networks.

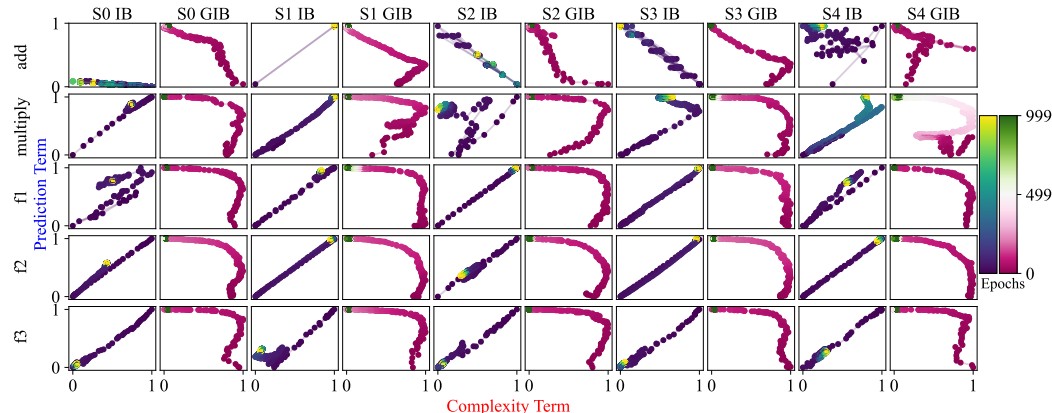

Figure 4: Information plane dynamics for NNs learning simple mathematical functions. Comparison of standard IB versus GIB across five functions (rows) and five random seeds. Functions include basic arithmetic and symmetric polynomials. GIB consistently shows compression phases (leftward movement), while standard IB exhibits varied behaviors. See Appendix E.3 for experimental details.

## 5.2 Information Dynamics of ResNets

We analyze information dynamics in residual networks (ResNets) of varying depths (20, 56, 80, 110 layers) trained on CIFAR-10. For the standard IB, we compute MI using the 10-dimensional output layer directly. For GIB, due to the need to compute feature-wise decompositions on the high-dimensional input space (3072 dimensions), we first apply Kernel PCA to reduce the pixel space to 50 principal components before computing MI, as explained in more detail in Appendix E.4 (Turk & Pentland, 1991). For more details as to why we chose 50 components, see Appendix H.

Figure 5 displays information plane trajectories for ResNets trained on CIFAR-10. The GIB formulation shows consistent compression and fitting for all network depths and random seeds, though the dynamics vary with architecture size. In smaller networks (ResNet-20), trajectories show a general trend of increasing prediction term while the complexity term decreases throughout training. Larger networks (ResNet-56 and above) begin to show phase structure.

The standard IB presents markedly different dynamics. Rather than showing clear phases, IB trajectories remain largely clustered with minimal compression across epochs. The absence of compression phases in standard IB holds across all tested architectures, confirming previous observations that $ReLU$ networks fail to exhibit expected IB behavior (Saxe et al., 2018).

## 5.3 Information Dynamics of Transformers Classifying News Headlines

We examine BERT-base fine-tuned on AG News text classification, comparing standard fine-tuning with a novel "unlearning" initialization strategy. In this case, unlearning involves training on random labels to remove biases from the model. For the standard IB, we again set $\mathcal{T}$ as the final layer representation for use in MI calculations. For GIB, we apply our feature-wise synergy decomposition to the full set of inputs. Full experimental details are in Appendix E.5.

The standard fine-tuning approach shown in row 1 of Figure 6 produces highly variable trajectories that begin near the center of the information plane, indicating that pre-trained BERT representations contain substantial pre-training biases. This prompted us to unlearn, where we train on random labels. The unlearning intervention dramatically alters these dynamics. After 3 epochs of random label training, models consistently start from the bottom-right corner of the information plane, as shown in row 2. From this reset position, both IB and GIB show more coherent learning trajectories during subsequent fine-tuning. This

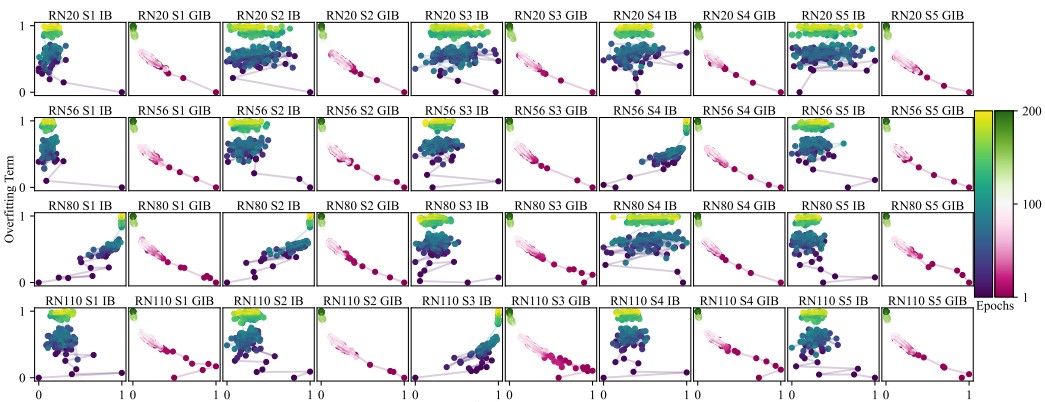

Figure 5: Information plane dynamics for ResNets of varying depths trained on CIFAR-10. Comparison across four network depths and five random seeds. GIB consistently exhibits compression phases, while standard IB shows limited or no compression. See Appendix E.4 for details.

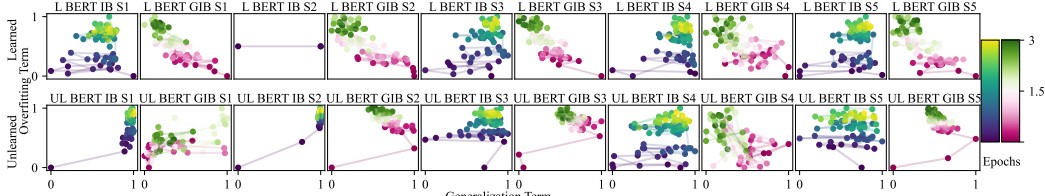

Figure 6: Information plane dynamics for BERT fine-tuned on AG News. Comparison of standard fine-tuning (top) versus unlearning + fine-tuning (bottom). The unlearning procedure repositions models to a more favorable initialization point for subsequent learning. See Appendix E.5 for details.

reveals how studying these information-planes can be used for diagnostics and interpretation.

### 5.4 Adversarial Robustness

We investigate how adversarial perturbations affect information dynamics by training NNs with *tanh* activations on MNIST under Fast Gradient Sign Method (FGSM) attacks of varying strength. Full details are in Appendix E.6. Figure 7(a) illustrates the effect of adversarial attacks on learning dynamics. Networks trained under weak attacks ($\epsilon = 0.01$) exhibit normal convergence, whereas strong attacks ($\epsilon = 1.0$) substantially hinder the learning process. The information-theoretic analysis in Figure 7(b) exposes a critical difference between standard IB and our GIB formulation. The GIB's complexity term faithfully reflects the degradation in generalization: values remain high for $\epsilon = 1.0$ (poor generalization), while decreasing rapidly when proper training occurs. In contrast, the standard IB's complexity term shows minimal differentiation between attack strengths.

## 6 Conclusion

In this work, we have introduced the Generalized Information Bottleneck (GIB), a principled reformulation of the IB that explicitly accounts for synergistic interactions between features. Our theoretical and empirical analysis demonstrated that synergistic functions achieve better generalization, providing a fundamental justification for why deep networks should learn representations that combine inputs synergistically rather than processing them independently. The GIB framework addresses several longstanding limitations of the original IB. First, we proposed a PMI-based reweighting scheme $Q(Z, Y)$ that protects the compression term from becoming infinite during training. Second, we have introduced a feature-wise synergy decomposition, explicitly penalizing representations that rely too heavily on individual

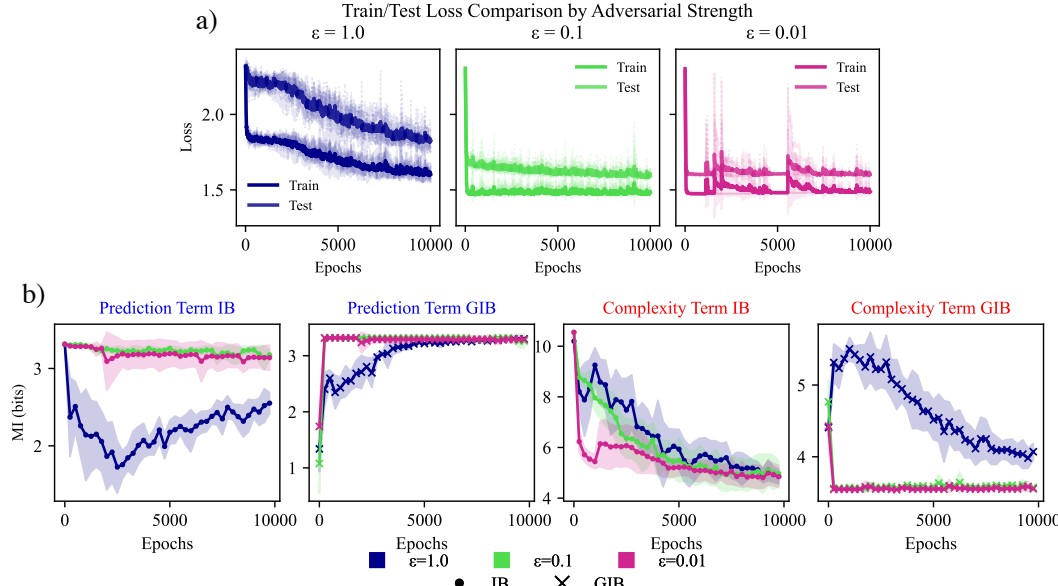

Figure 7: Information dynamics under adversarial attacks on MNIST. (a) Training dynamics for three FGSM attack strengths. (b) Information-theoretic analysis comparing IB versus GIB. GIB's complexity term correctly reflects degradation in generalization, while standard IB fails to differentiate between attack strengths. See Appendix E.6 for details.

features or simple feature combinations. This ensures that our formulation highlights when networks learn patterns that emerge solely through the collective processing of multiple inputs, a distinction that is crucial for robust generalization. Our experimental results across diverse architectures demonstrate that GIB provides a more complete picture of how deep networks process information. The GIB framework opens new possibilities for both understanding and improving deep learning systems.

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

## A    Notation Table

Table 1 contains all the mathematical notation used in this paper.

Table 1: Summary of Notational Conventions.

| Symbol | Description |
| --- | --- |
| $\mathcal{X}$ | Set of input random variables (features) |
| $X^i$ | $i$-th individual input feature |
| $\mathcal{X}^{-i}$ | All features except the $i$-th feature: $\mathcal{X} \setminus \{X^i\}$ |
| $Y$ | Target random variable (labels) |
| $Z$ | Output/prediction random variable |
| $\mathcal{T}$ | Hidden representation/latent space |
| $\varepsilon$ | Noise random variable |
| $N$ | Number of input features |
| $\beta$ | Trade-off parameter in IB formulation |
| $I(\cdot; \cdot)$ | Mutual information |
| $H(\cdot)$ | Entropy |
| $H(\cdot\|\cdot)$ | Conditional entropy |
| $P(\cdot)$ | Probability distribution |
| $Q(Z, Y)$ | PMI-based reweighted distribution |
| $Q$ | Variable sampled from $Q(Z, Y)$ |
| $\mathrm{Syn}(\cdot)$ | Synergy measure |
| $\mathrm{s}^+, \mathrm{s}^-$ | More/less synergistic functions |
| IB | Information Bottleneck |
| GIB | Generalized Information Bottleneck |
| MI | Mutual Information |
| NN | Neural Network |
| PMI | Point-wise Mutual Information |
| PID | Partial Information Decomposition |
| FGSM | Fast Gradient Sign Method |

## B    Proof of Theorem 1

In this section we prove Theorem 1. To do this, we assume that we have perfect training performance and therefore $Q(Y, Z) = Y = Z$. We also assume the predictor is deterministic given its input (as in a standard feed-forward network), hence $H(Z|\mathcal{X}) = 0$ and therefore

$$I(\mathcal{X}; Z) = H(Z). \tag{7}$$

*Proof.* Given this, our GIB formulation becomes:

$$\mathcal{L}_{\text{GIB}} = I(Z;Y) - \frac{1}{2\beta N} \sum_{i=1}^{N} \left( I(\mathcal{X}^{-i};Q) + I(X^i;Q) \right) \tag{8}$$

$$= H(Y) - \frac{1}{2\beta N} \sum_{i=1}^{N} \left( I(\mathcal{X}^{-i};Z) + I(X^i;Z) \right) \quad \text{(since } Z = Y \text{ and } Q = Z = Y) \tag{9}$$

$$\geq I(\mathcal{X};Y) - \frac{1}{2\beta N} \sum_{i=1}^{N} \left( I(\mathcal{X}^{-i};Z) + I(X^i;Z) \right) \quad \text{(since } I(\mathcal{X};Y) \leq H(Y)) \tag{10}$$

$$\geq I(\mathcal{X};Y) - \frac{1}{2\beta N} \sum_{i=1}^{N} \left( H(Z) + H(Z) \right) \quad \text{(by } I(A;B) \leq H(B)) \tag{11}$$

$$= I(\mathcal{X};Y) - \frac{1}{2\beta N} \cdot N \cdot 2H(Z) \tag{12}$$

$$= I(\mathcal{X};Y) - \frac{1}{\beta} H(Z) \tag{13}$$

$$= I(\mathcal{X};Y) - \frac{1}{\beta} I(\mathcal{X};Z) \quad \text{(by Eq. 7, } H(Z|\mathcal{X}) = 0) \tag{14}$$

$$\geq I(\mathcal{X};Y) - \frac{1}{\beta} I(\mathcal{X};\mathcal{T}) \quad \text{(by data processing, } \mathcal{X} \to \mathcal{T} \to Z) \tag{15}$$

$$\geq I(\mathcal{T};Y) - \frac{1}{\beta} I(\mathcal{X};\mathcal{T}) \quad \text{(by data processing, } \mathcal{X} \to \mathcal{T} \to Y). \tag{16}$$

We obtain:

$$I(\mathcal{T};Y) - \frac{1}{\beta} I(\mathcal{X};\mathcal{T}) \leq \mathcal{L}_{\text{GIB}}$$

which shows that the original IB objective is upper bounded by the proposed GIB under the stated assumption. $\qquad\square$

## C  GIB AND SUFFICIENT STATISTICS

In this section, we show that when the first term in the GIB objective is $I(Z;Y)$, the GIB recovers sufficient statistics in the limit $\beta \to \infty$.

**Theorem 2.** *For $\beta \in (0, \infty]$, consider the GIB functional*

$$\mathcal{J}_\beta(P_{Z|X}) := I(Z;Y) - \frac{1}{\beta\,2N} \sum_{i=1}^{N} \left( I(X^{-i};Q) + I(X^i;Q) \right).$$

*Then at $\beta = \infty$,*

$$\sup_{P_{Z|X}} \mathcal{J}_\infty = \sup_{P_{Z|X}} I(Z;Y) \ \leq \ H(Y).$$

*Moreover, equality holds (i.e., $\sup_{P_{Z|X}} I(Z;Y) = H(Y)$) if and only if $Y$ is a deterministic function of $Z$, equivalently $H(Y \mid Z) = 0$. In this case, $Z$ is a sufficient statistic.*

*Proof.* At $\beta = \infty$, the penalty term vanishes and therefore $\mathcal{J}_\infty = I(Z;Y)$. By the elementary information inequality $I(Z;Y) \leq H(Y)$ for all encoders $P_{Z|X}$, we have

$$\sup_{P_{Z|X}} \mathcal{J}_\infty = \sup_{P_{Z|X}} I(Z;Y) \leq H(Y).$$

If for some encoder we have $H(Y \mid Z) = 0$, i.e. $Y$ is a deterministic function of $Z$, then

$$I(Z;Y) = H(Y) - H(Y \mid Z) = H(Y),$$

and the upper bound is achieved.

Conversely, if $I(Z;Y) = H(Y)$ for some encoder, then

$$H(Y \mid Z) = H(Y) - I(Z;Y) = 0,$$

so $Y$ must be a deterministic function of $Z$.

Thus, at $\beta = \infty$, maximising $\mathcal{J}_\infty$ recovers precisely the set of encoders for which $Z$ is a sufficient statistic for $Y$. □

## D   WHEN GIB ENCOUNTERS INFINITY

In this section, we analyze the conditions under which our GIB formulation yields infinite values and show that, unlike standard IB, these infinities have meaningful interpretations.

**Theorem 3.** *Let $\mathcal{X} = (X^1, \ldots, X^N)$ be a feature vector and let $Y$ be a continuous random variable. Assume that the representation $Z$ achieves perfect prediction of $Y$ in the sense that $I(Z;Y) = \infty$. Assume further that no individual feature $X^i$ nor its complement $\mathcal{X}^{-i}$ alone yields an infinite mutual information with the PMI variable $Q$, in the sense that*

$$\frac{1}{N} \sum_{i=1}^{N} \left( I(\mathcal{X}^{-i}; Q) + I(X^i; Q) \right) < \infty,$$

*where $Q$ is the point-wise mutual information random variable. Then $\mathcal{L}_{\text{GIB}} = \infty$.*

*Proof.* By definition,

$$\mathcal{L}_{\text{GIB}} = I(Z;Y) - \frac{1}{2\beta N} \sum_{i=1}^{N} \left( I(\mathcal{X}^{-i}; Q) + I(X^i; Q) \right).$$

By assumption, $I(Z;Y) = \infty$, while the sum is finite. Hence, in the extended real line,

$$\mathcal{L}_{\text{GIB}} = \infty - (\text{finite}) = \infty.$$

□

**Interpretation of Infinities in GIB.**   In contrast to the standard IB, where infinities arise as technical artifacts of deterministic mappings between continuous variables, the infinities in the GIB functional admit a more structural interpretation.

When $\mathcal{L}_{\text{GIB}} = \infty$, we are in a regime of *perfect synergy*: the output $Y$ can be recovered from the full feature set (via $Z$), yet no individual coordinate or coordinate subset carries enough information to reconstruct the relevant PMI signal. In other words, all features are jointly essential. This corresponds to the ideal synergistic regime.

Conversely, if some feature $X^i$ or complement $\mathcal{X}^{-i}$ can alone perfectly determine $Y$, then the corresponding term $I(X^i; Q)$ or $I(\mathcal{X}^{-i}; Q)$ becomes infinite, making the entire penalty infinite. In this case, the GIB takes the form

$$\infty - \infty,$$

which is *indeterminate*. This suggests that prediction is not synergistic, as the output can be reconstructed from a sufficiently informative subset of the features.

Thus, while both IB and GIB encounter infinities in the continuous setting, the GIB admits a meaningful interpretation: an infinite value corresponds precisely to perfect synergy, whereas the indeterminate case reflects the absence of synergy.

## E   EXPERIMENTAL SETTINGS

### E.1   SYNTHETIC SYNERGY EXPERIMENT

**Data Generation.**   For each input dimension $n \in \{2, ..., 10\}$, we generate $N = 10^6$ samples. Each sample consists of a binary input vector $X \in \{0,1\}^n$ with i.i.d. Bernoulli(0.5) entries. We apply a "force-to-1" noise model: with probability $p_{\text{flip}} = 1/3$, we randomly select one coordinate $i \sim \text{Uniform}\{1, ..., n\}$ and set $X'_i = 1$, leaving all other coordinates unchanged. With probability $2/3$, no modification is made ($X' = X$). The noise pattern is encoded as $\varepsilon \in \{0, 1, ..., n\}$, where 0 indicates no flip and $i > 0$ indicates coordinate $i$ was forced to 1.

**Functions.** We examine three deterministic functions of increasing synergy applied to the noisy input $X'$:

- Non-synergistic: $f_1(X') = X_1'$ (output depends only on first input)
- Partially synergistic: $f_2(X') = X_1' \oplus X_2'$ (XOR of first two inputs)
- Highly synergistic: $f_3(X') = \bigoplus_{i=1}^{n} X_i'$ (XOR of all inputs)

**MI Estimation.** Since all variables are discrete, we compute exact MI using empirical probability distributions with base-2 logarithms.

### E.2 CIFAR-10 Synergy with Augmentation

**Architecture and Training.** We train ResNet models of depths {20, 32, 44, 56, 68, 80, 92, 110} on CIFAR-10. Each architecture follows the standard ResNet design for CIFAR with initial $3 \times 3$ convolution, three residual stages, global average pooling, and a final 10-way linear classifier. Models are trained with SGD (learning rate 0.1, momentum 0.9, weight decay $5 \times 10^{-4}$) for 200 epochs with batch size 256. Learning rate is reduced by a factor of 0.1 at epochs 100 and 150 using MultiStepLR scheduler. Standard data augmentation consists of random crops ($32 \times 32$ with padding 4) and horizontal flips applied during training.

**Teacher-Student Framework.** To quantify synergy with augmentation noise, we employ a two-stage approach. First, a teacher network is trained as described above on augmented data. After training, we collect the teacher's softmax outputs on the augmented training set. We then train a student network of identical architecture to predict these teacher outputs from non-augmented inputs. The student is trained for 200 epochs using the same SGD configuration (lr=0.1, momentum=0.9, weight decay=$5 \times 10^{-4}$) with MultiStepLR milestones at epochs 100 and 150. The student minimizes cross-entropy loss between its predictions and the teacher's softmax targets.

**Synergy Measurement.** We compute the marginal entropy of teacher predictions as $H(Y) = -\mathbb{E}[(p_{\text{teacher}} \log p_{\text{teacher}})]$ where the expectation is over all augmented training samples. The conditional entropy is estimated as the final cross-entropy loss achieved by the converged student model. The synergy ratio $I(f(X, \varepsilon); \varepsilon | X) / I(f(X, \varepsilon); X, \varepsilon)$ (which can be re-written as the final loss of the student divided by the total entropy) quantifies the proportion of the teacher's output entropy that cannot be predicted from clean images alone, requiring knowledge of the augmentation pattern.

### E.3 Simple Functions

**Network Architecture.** All networks consist of a single hidden layer with specified units, followed by a linear output layer. No bias terms, regularization, or normalization are used. Weights are initialized using PyTorch's default settings, namely Kaiming uniform for the hidden layers and uniform initialization for the output layer.

**Target Functions and Architectures.** The target functions and architectures considered in our evaluation are the following:

- Addition: $f(a, b) = a + b$, 2 inputs → 4 hidden units (identity activation) → 1 output;
- Multiplication: $f(a, b) = a \times b$, 2 inputs → 3 hidden units (square activation: $x^2$) → 1 output;
- Symmetric polynomial 1 (f1): $f(a, b, c) = ab + bc + ca$, 3 inputs → 16 hidden units (square activation: $x^2$) → 1 output;
- Symmetric polynomial 2 (f2): $f(a, b, c) = a^2 + b^2 + c^2$, 3 inputs → 8 hidden units (square activation: $x^2$) → 1 output;
- Symmetric polynomial 3 (f3): $f(a, b, c, d) = ab + bc + cd + da$, 4 inputs → 16 hidden units (square activation: $x^2$) → 1 output.

**Training Details.** Networks are trained with standard gradient descent (no momentum) with learning rate 0.01 for 1000 epochs, minimizing mean squared error (MSE) loss. Training data consists of 1500 samples uniformly sampled from $[-10, 10]^n$ for all functions except addition, which uses $[0, 10]^2$. Test data uses 1500 samples from the extended range $[-1000, 1000]^n$ to evaluate extrapolation. MI is computed every 10 epochs using histogram binning with 40 bins. We get our binning estimation technique from Saxe et al. (2018).

### E.4 RESNETS ON CIFAR-10

**Architecture.** We use standard ResNet architectures for CIFAR-10 with depths $\{20, 56, 80, 110\}$, implemented with BasicBlocks. Each network has an initial 16-filter $3 \times 3$ convolution, followed by three stages with $\{16, 32, 64\}$ filters respectively, global average pooling, and a 10-way linear classifier.

**Training.** Models are trained on CIFAR-10 (50k train, 10k test) with SGD (momentum 0.9, weight decay $5 \times 10^{-4}$, initial learning rate 0.1) for 200 epochs with batch size 128. Learning rate decays by 0.1 at epochs 100 and 150. Standard augmentation includes random crops ($32 \times 32$, padding 4) and horizontal flips. No additional preprocessing is applied beyond standard CIFAR normalization.

**MI Estimation.** For standard IB, we compute MI between the 10-dimensional logit vector $\mathcal{T}$ (pre-softmax outputs) and targets $Y$. For GIB's input decomposition, we first apply Kernel PCA with RBF kernel (gamma=1/3072) to reduce the 3072-dimensional flattened images to 50 principal components. MI is computed at each epoch using the first 5000 training samples with histogram binning (30 bins). For IB: $I(\mathcal{T}; Y)$ using the 10-dimensional logits. For GIB: synergy decomposition using the 50 PCA components as features.

### E.5 BERT ON AG NEWS

**Model Configuration.** BERT-base-uncased (12 layers, 768 hidden dimensions, 12 attention heads) fine-tuned for 4-way AG News classification (World, Sports, Business, Sci/Tech). The dataset contains 120,000 training and 7,600 test examples. Maximum sequence length is 128 tokens with padding.

**Training Protocols.** The training protocols used in our evaluation are the following:

- **Standard Fine-tuning:** Direct fine-tuning from pre-trained BERT weights for 3 epochs.
- **Unlearning + Fine-tuning:** 3 epochs of training with randomly shuffled labels (maintaining class balance), followed by 3 epochs of standard fine-tuning.

**Optimization.** Both protocols use AdamW optimizer with learning rate $2 \times 10^{-5}$ and weight decay 0.01, batch size 32. No learning rate warmup or scheduling is applied. Training uses cross-entropy loss over the 4 classes.

**MI Computation.** MI is computed 24 times per epoch (approximately every 200 batches) using 5000 training samples. For standard IB, we use the 4-dimensional logit vector $\mathcal{T}$ from the classification head. For GIB, we use the raw 128-dimensional token ID sequences as input features $\mathcal{X}$ (no PCA is applied). MI estimation uses histogram binning with 30 bins.

### E.6 ADVERSARIAL ROBUSTNESS

**Architecture.** 4-layer fully-connected network: $784 \to 1024 \to 20 \to 20 \to 20 \to 10$, with *tanh* activations after each hidden layer and softmax output.

**Adversarial Training.** FGSM attacks are applied to every training example in each batch: $x_{adv} = x + \epsilon \cdot \text{sign}(\nabla_x \mathcal{L}(f(x), y))$ where $\epsilon \in \{0.01, 0.1, 1.0\}$. Perturbed inputs are clipped to

[0,1]. The training loss is the average of clean and adversarial losses: $\mathcal{L} = (\mathcal{L}_{clean} + \mathcal{L}_{adv})/2$. No validation set or early stopping is used.

**Training Details.** Networks are trained for 10,000 epochs using Adam optimizer with learning rate $10^{-3}$. MI is computed every 250 epochs between inputs and the final 20-dimensional hidden layer activations using histogram binning (30 bins).

## F  ALTERNATIVE MI ESTIMATION USING GCMI AND KDE

To validate the robustness of our findings, we repeated our experiments using two alternative MI estimation methods: Gaussian Copula Mutual Information (GCMI) and Kernel Density Estimation (KDE). These methods offer different trade-offs between computational efficiency and estimation accuracy compared to our primary histogram binning approach.

### F.1  KERNEL DENSITY ESTIMATION (KDE)

KDE (Parzen, 1962) estimates probability densities using kernel functions centered at each data point, then computes MI from these continuous density estimates. We use Gaussian kernels with bandwidth selected via Scott's rule (Scott, 1992). While computationally more intensive than GCMI ($\mathcal{O}(n^2)$ for $n$ samples), KDE provides non-parametric estimates that can capture arbitrary distribution shapes without assuming specific parametric forms. This flexibility makes KDE particularly suitable for complex, multi-modal distributions that might arise in neural network representations.

### F.1.1  ACTIVATION FUNCTION COMPARISON

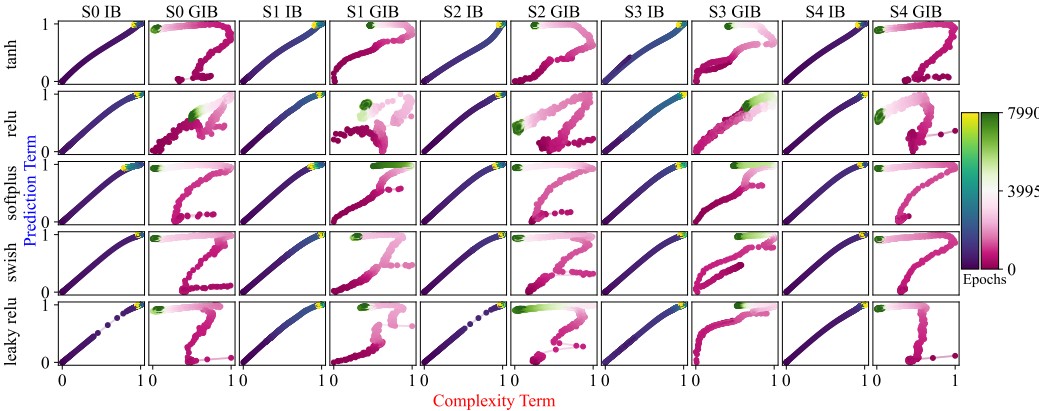

Figure 8: Information plane dynamics across activation functions using KDE estimation. The non-parametric nature of KDE reveals fine-grained dynamics in the information plane trajectories.

Figure 8 displays information plane dynamics using KDE estimation. The GIB formulation (pink) again shows compression phases across all activation functions. Standard IB (blue) fails to show any compression.

### F.1.2  SIMPLE MATHEMATICAL FUNCTIONS

Figure 9 shows KDE-based MI estimation for simple function learning. The GIB formulation exhibits clear compression phases for all functions. Although, this can also be accompanied by a stage of decompression. Standard IB shows more erratic behavior, generally moves upward and leftward but in less distinct phases.

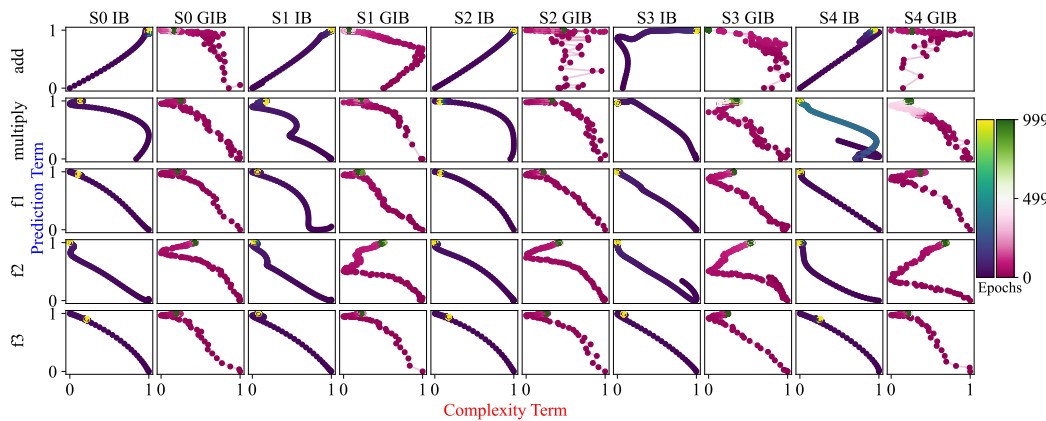

Figure 9: Information plane dynamics for simple mathematical functions using KDE estimation. The method provides sharp phase transitions despite increased trajectory variance.

### F.1.3    ResNet Information Dynamics

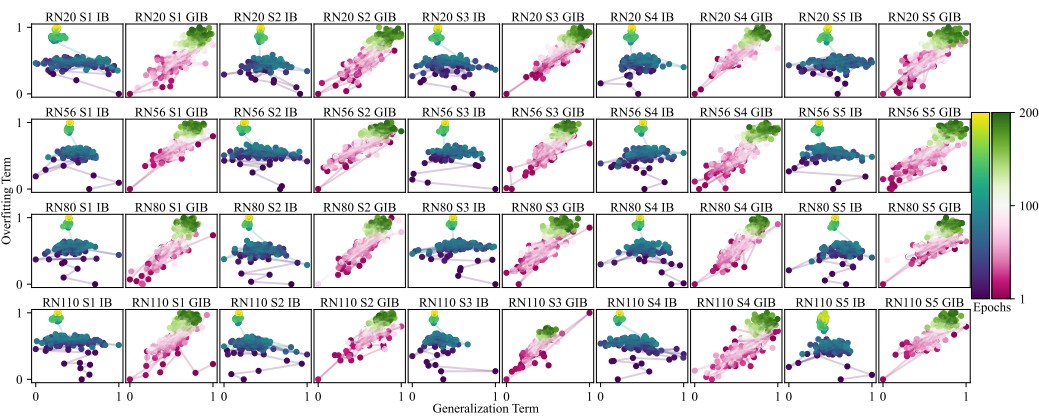

Figure 10: Information plane dynamics for ResNets on CIFAR-10 using KDE estimation.

Figure 10 reveals the first significant divergence in interpretation between the GIB results obtained using binning versus KDE. Under KDE estimation, neither GIB nor IB exhibits interpretable information bottleneck dynamics.

### F.1.4    BERT Fine-tuning Dynamics

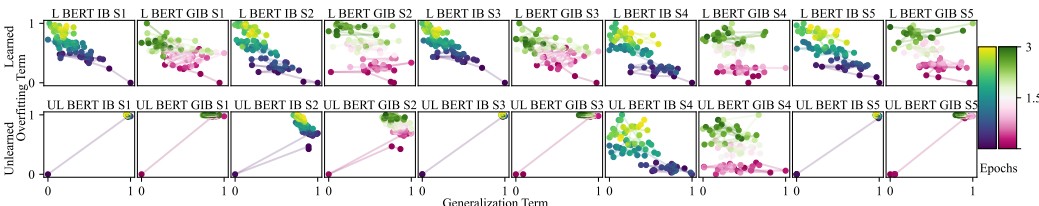

Figure 11: Information plane dynamics for BERT fine-tuning using KDE estimation.

In Figure 11, our results realign with the interpretation presented in the main text. Following the unlearning phase, we observe a rapid initial fitting step succeeded by a prolonged, gradual (and less pronounced) compression phase. These dynamics are evident for both IB and GIB.

## F.2 GAUSSIAN COPULA MUTUAL INFORMATION (GCMI)

GCMI (Ince et al., 2017) estimates MI by first transforming variables to have standard Gaussian marginals using the Gaussian copula, then computing MI under the Gaussian assumption. This approach is particularly effective for continuous variables with complex, potentially non-linear relationships. The method is computationally efficient (O(n log n) for n samples) and provides robust estimates even for high-dimensional data. Unlike histogram binning, GCMI does not require discretization parameters and automatically adapts to the data distribution.

### F.2.1 ACTIVATION FUNCTION COMPARISON

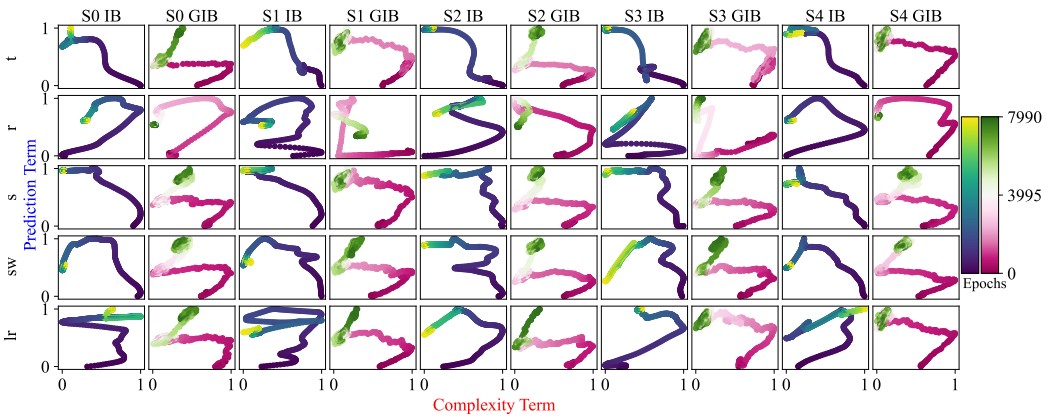

Figure 12: Information plane dynamics across activation functions using GCMI estimation. Standard IB (blue) and GIB (pink) trajectories for networks trained on synthetic tasks S0-S4.

Figure 12 presents information plane dynamics using GCMI estimation across five activation functions. Under this estimator, both methods frequently exhibit compression phases. However, GIB often displays a subsequent decompression phase, characterized by a rightward shift in the later stages of training.

### F.2.2 SIMPLE MATHEMATICAL FUNCTIONS

Figure 13 presents results for networks learning arithmetic and polynomial functions using GCMI estimation. While the dynamics of both methods are not easily interpretable, the GIB formulation demonstrates upward leftward movement, indicating the occurrence of fitting and compression but they are occurring at once rather than in distinct phases. In contrast, standard IB shows variable behavior with limited evidence of compression.

### F.2.3 RESNET INFORMATION DYNAMICS

In Figure 14, we observe that neither the IB nor the GIB yields interpretable results.

### F.2.4 BERT FINE-TUNING DYNAMICS

Figure 15 likewise shows that the dynamics produced by both the IB and GIB are not interpretable.

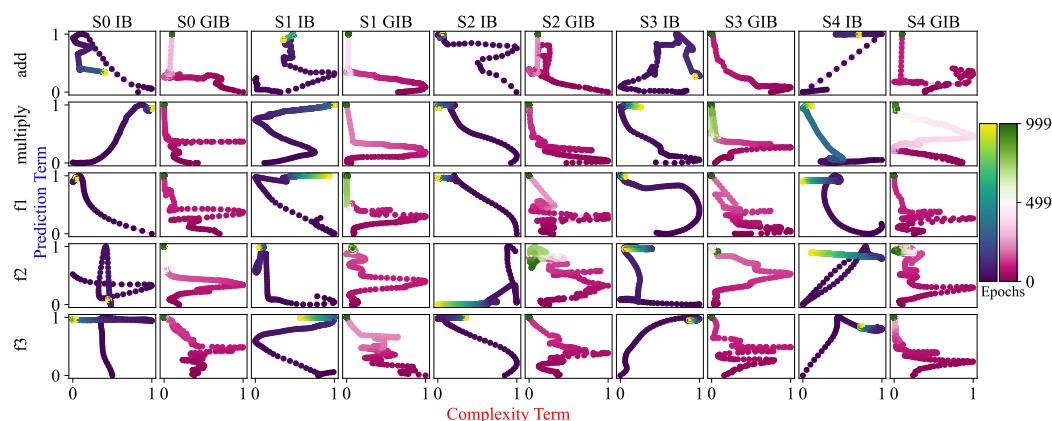

Figure 13: Information plane dynamics for simple mathematical functions using GCMI estimation.

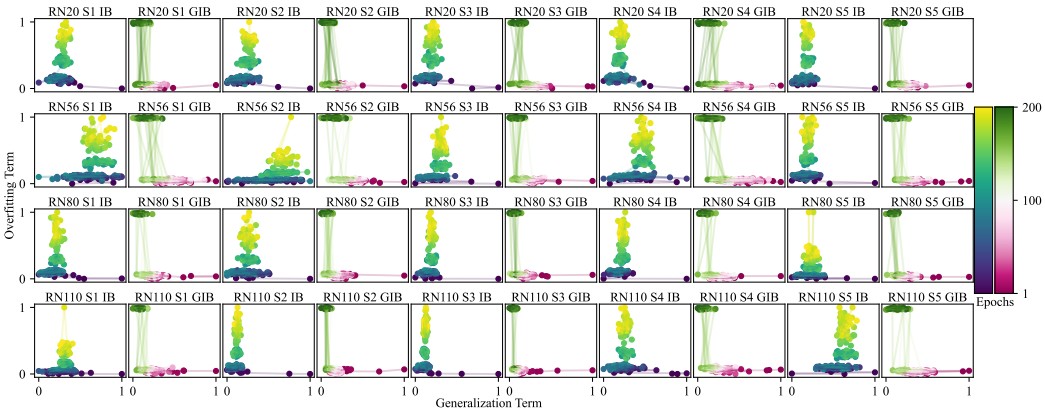

Figure 14: Information plane dynamics for ResNets on CIFAR-10 using GCMI estimation.

### F.3  SUMMARY OF GCMI AND KDE RESULTS

In this section, KDE provides strong validation for our main findings. For the KDE method, our GIB formulation consistently shows compression phases where standard IB fails, particularly for different activation functions and simple arithmetic. Meanwhile, the effect is less pronounced for GCMI, but compression remains more likely than for the standard IB. This consistency across three fundamentally different MI estimation approaches (binning, GCMI, and KDE) strongly supports our theoretical framework. The fact that synergy-based decomposition reveals consistent information dynamics across estimation methods suggests that GIB captures a fundamental aspect of how neural networks process information during learning.

## G  COMPUTATIONAL COMPLEXITY

The computational requirements of GIB and IB differ significantly in their scaling behavior. If we define our unit of computation as a single MI estimation, GIB requires $2N + 1$ calculations: more specifically, one for the prediction term $I(Z; Y)$ and $2N$ for the complexity term (computing $I(\mathcal{X}^{-i}; Q)$ and $I(X^i; Q)$ for each feature). Critically, these calculations occur at the input layer where dimensionality is typically highest, for CIFAR-10, this means 3072

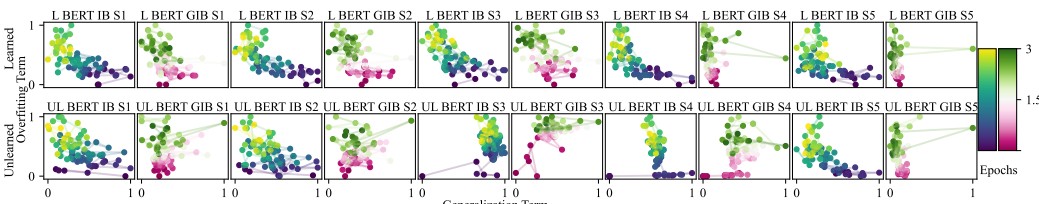

Figure 15: Information plane dynamics for BERT fine-tuning using GCMI estimation.

features. However, GIB's computational cost is independent of network depth, since it only considers input-output relationships. In contrast, standard IB requires $2L$ MI calculations for $L$ layers, computing $I(\mathcal{X};\mathcal{T}_l)$ and $I(\mathcal{T}_l;Y)$ at each layer. While one might compute IB only for the final layer where dynamics are often most pronounced, this prevents the use of the IB as a tool for understanding learning dynamics throughout the network. Additionally, GIB benefits from a key advantage: we can apply PCA to high-dimensional inputs (as we do for CIFAR-10 in Section 5.2) because features at the input layer share a common representation space (Turk & Pentland, 1991). Conversely, combining representations across layers for IB is less conventional.

## H   EFFECT OF PCA DIMENSIONALITY ON GIB DYNAMICS

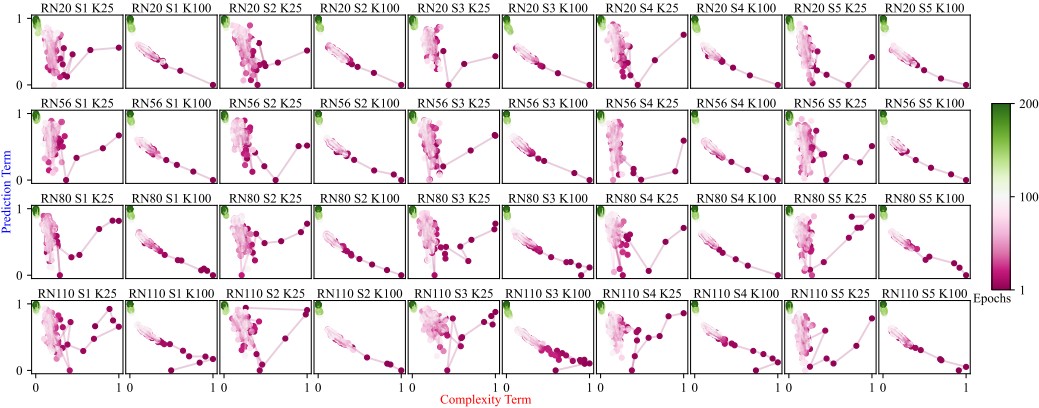

Figure 16: GIB information plane dynamics for ResNets with varying PCA dimensionality. Each subplot shows results for KPCA with 25 (left) versus 100 (right) components (C's). As dimensionality increases, compression phases become more pronounced.

Figure 16 demonstrates the impact of PCA dimensionality on observed GIB dynamics. With only 25 principal components, the dynamics are erratic and noisy. However, as we increase to 50 components (shown in main results) and then to 100 components, the dynamics become less noisy. This progression suggests that capturing synergistic information requires sufficient dimensionality to represent the complex feature interactions present in the original input space.

# I    Sum versus Whole Synergy Formulation

## I.1    Sum-Versus-Whole Synergy

In this section, we examine an alternative formulation of synergy based on sum versus whole synergy rather than our feature-wise approach. Due to the increased noise in this estimation method, all MI values are averaged over 50 iterations to obtain stable measurements.

This alternative definition of GIB is based on the sum-versus-whole formulation of synergy, which compares information available from the complete feature set against the sum of information from individual components (Schneidman et al., 2003). The basic form is $\mathrm{Syn}_{\mathrm{GIB}}(\mathcal{X} \to Y) = I(\mathcal{X}; Y) - \sum_{i=1}^{N}\big(I(X^i; Y)\big)$, which considers only individual features. This captures the intuitive notion of synergy, for example, XOR has zero information from individual inputs but perfect information from their combination, yielding maximal synergy (Bell, 2003). Unlike exponentially complex PID-based measures (Williams & Beer, 2010), this formulation requires only $O(N)$ mutual information calculations, making it computationally feasible for tracking synergistic learning dynamics in high-dimensional neural networks . Combining this with our representation of the PMI-weighted combination of $Z$ and $Y$ and rewriting as a Lagrangian optimization we get the following:

$$\mathcal{L}_{\mathrm{SVW}} = \max_{p(Z|X)} \left[ \underbrace{I(Z;Y)}_{\text{prediction term}} - \beta^{-1} \underbrace{\sum_{i=1}^{N} I(X^i; Q)}_{\text{complexity term}} \right] \tag{17}$$

In the following section, we compare the outcomes of tracking this optimization with those obtained from the approach introduced in the main paper.

## I.2    Activation Function Comparison

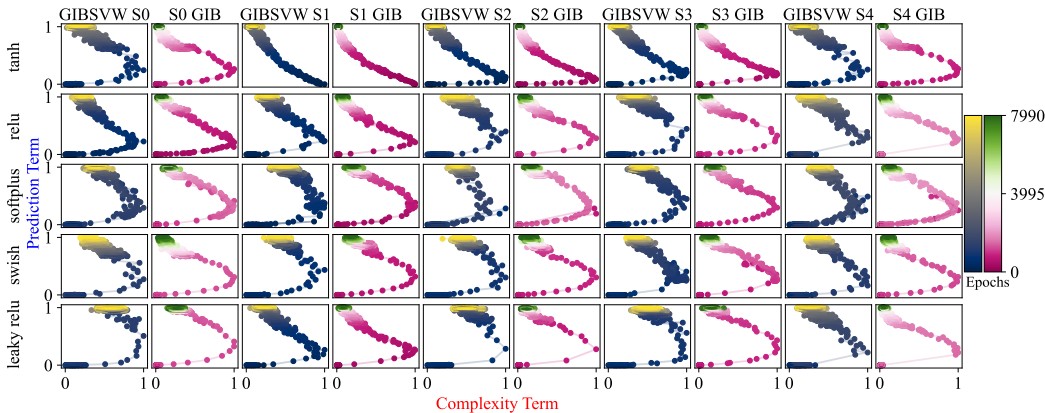

Figure 17: Information plane dynamics comparing the alternative synergy bottleneck (SVW, blue) with our GIB (pink) across multiple activation functions. While SVW shows compression phases more frequently than standard IB, it exhibits less consistent compression than our feature-wise GIB formulation. MI values averaged over 50 iterations.

Figure 17 shows that the alternative synergy formulation (SVW) improves upon standard IB by exhibiting compression phases in several cases where IB fails. However, the compression is less pronounced and less consistent across activation functions compared to our feature-wise GIB. This suggests that while any synergy-based decomposition provides benefits over treating the latent space as a black box, the specific choice of synergy might impact the observability of information dynamics.

### I.3   Simple Mathematical Functions

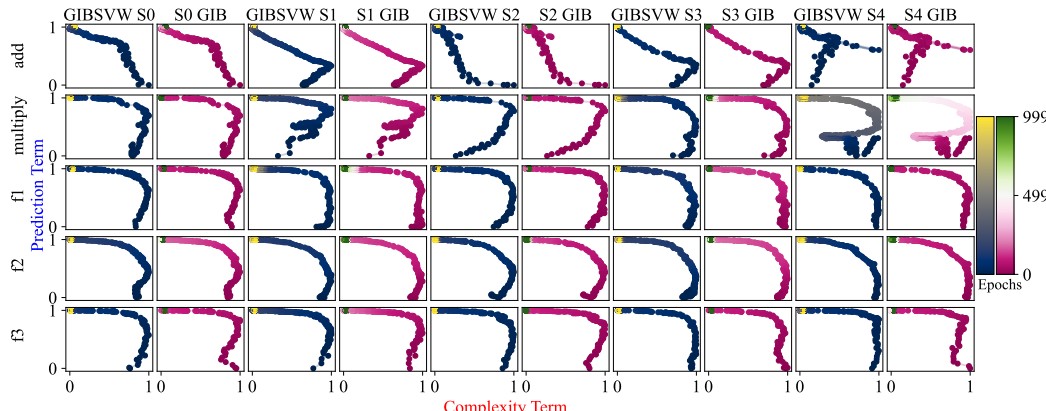

Figure 18: Learning dynamics for simple mathematical functions. The alternative synergy bottleneck (SVW) consistently shows compression phases across all functions, significantly outperforming standard IB.

For NNs learning simple mathematical functions (Figure 18), the alternative synergy formulation consistently exhibits compression phases across all tasks. This represents a substantial improvement over standard IB, which shows no compression for these functions.

### I.4   ResNet Information Dynamics

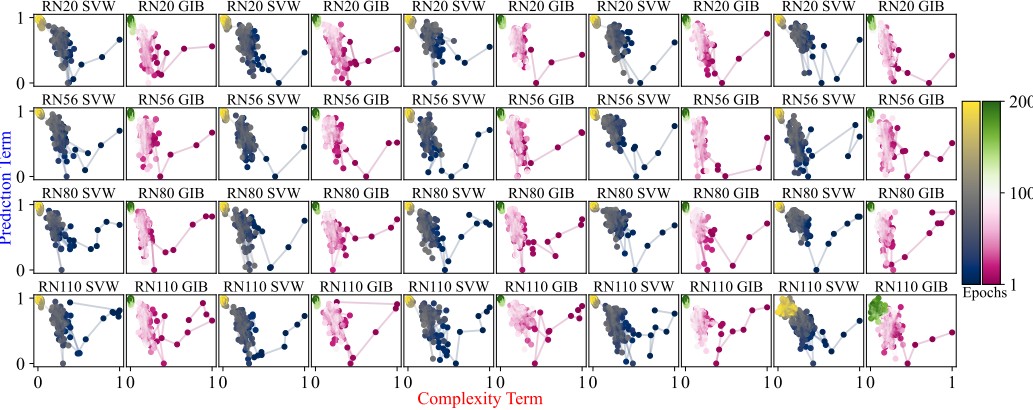

Figure 19: ResNet information dynamics on CIFAR-10. The alternative synergy bottleneck (SVW) shows distinct compression and non-compression phases, providing clearer learning dynamics than standard approaches while exhibiting more variability than our feature-wise GIB.

Figure 19 shows that the alternative synergy formulation reveals distinct phases in ResNet training.

### I.5   BERT Fine-tuning Dynamics

The first notable limitation of the alternative synergy formulation appears in transformer fine-tuning (Figure 20). The SVW method fails to exhibit compression phases for BERT on the AG News classification task, even after our unlearning intervention. In contrast, our GIB

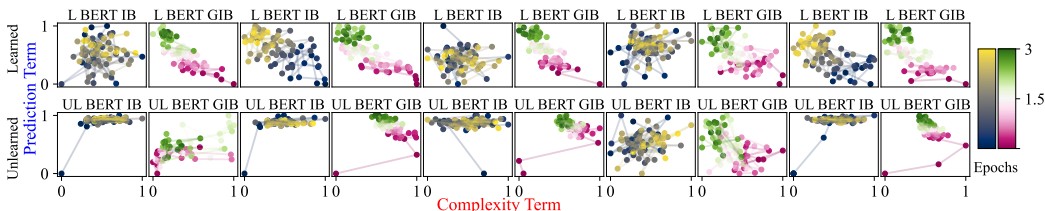

Figure 20: BERT fine-tuning with the alternative synergy bottleneck. Unlike our GIB formulation, SVW fails to show compression phases for both standard fine-tuning and the unlearning protocol, suggesting inherent limitations in capturing synergistic dynamics in transformer architectures.

formulation clearly reveals compression dynamics. This discrepancy suggests that variance-weighted synergy measures may struggle to capture the high-dimensional, attention-based computations characteristic of transformers.

## I.6    SUMMARY

The alternative synergy formulation serves as a useful baseline, showing that synergy-based approaches generally outperform standard IB.

