# OpenReview forum: "A Generalized Information Bottleneck Theory of Deep Learning"
_ICLR.cc/2026/Conference — ICLR 2026 Conference Desk Rejected Submission_

### Official Review · Reviewer_ZKFn · 2025-10-27

**Soundness:** 2
**Presentation:** 2
**Contribution:** 2
**Rating:** 2
**Confidence:** 4

**Summary:**

This paper proposes a Generalized Information Bottleneck (GIB) framework that reformulates the original Information Bottleneck principle through the lens of synergy - information obtainable only through joint processing of features. The authors claim that synergistic functions achieve superior generalization and present a computable definition of synergy based on average interaction information.

**Strengths:**

1. GIB demonstrates consistency in exhibiting compression phases across diverse architectures and activation functions.
2. The complexity term's correlation with adversarial vulnerability provides a quantitative tool for analyzing model robustness.

**Weaknesses:**

1. The computational complexity of the GIB approach is prohibitive for real-world applications, requiring 2N+1 mutual information calculations for N features.

2. Mutual information estimation via binning is known to be inaccurate (Saxe,et.al,2019) yet all major conclusions rely on this flawed estimation method.

3. No comparison with other state-of-the-art generalization techniques (NIB [1] , HSIC [2], DIB [3], Gate-IB [4]) or information-theoretic approaches.

4.  Incorporates the concept of synergy lacks sufficient novelty, previous work already uses more accurate way (matrix-based Rényi's mutual information estimation techniques [5] [6]) to estimate the synergy. Building on these foundations, several established information bottleneck variants  (DIB) [3], Gate-IB [4], and Multi-view Information Bottleneck [7] have already incorporated synergistic principles into their formulations.

[1] Kolchinsky, et al. "Nonlinear information bottleneck." Entropy 21.12 (2019): 1181.

[2] Wang, Zifeng, et al. "Revisiting hilbert-schmidt information bottleneck for adversarial robustness." Advances in Neural Information Processing Systems 34 (2021): 586-597.

[3] Yu, Xi, et al. "Deep deterministic information bottleneck with matrix-based entropy functional." ICASSP 2021-2021 IEEE International Conference on Acoustics, Speech and Signal Processing (ICASSP). IEEE, 2021.

[4] Alesiani, Francesco, et al. "Gated information bottleneck for generalization in sequential environments." Knowledge and Information Systems 65.2 (2023): 683-705.

[5] Yu, Shujian, et al. "Multivariate Extension of Matrix-Based Rényi's $\alpha $ α-Order Entropy Functional." IEEE transactions on pattern analysis and machine intelligence 42.11 (2019): 2960-2966.

[6] Yu, Shujian, et al. "Measuring dependence with matrix-based entropy functional." Proceedings of the AAAI Conference on Artificial Intelligence. Vol. 35. No. 12. 2021.

[7] Zhang, Qi, et al. "Multi-view information bottleneck without variational approximation." ICASSP 2022-2022 IEEE International Conference on Acoustics, Speech and Signal Processing (ICASSP). IEEE, 2022.

**Questions:**

1. The paper demonstrates that GIB exhibits compression phases where standard IB fails, but does this translate to measurable improvements in test accuracy or other standard generalization metrics? Can you provide a quantitative analysis?

2. The PCA preprocessing used for CIFAR-10 experiments potentially discards important synergistic relationships. Have you validated that the reduced representation preserves the synergistic interactions you aim to measure?

---

> ### Author Response · Authors · 2025-11-27
> **Response**
>
> ---
>
> ### Comment 1
> **Reviewer:**
> The computational complexity of the GIB approach is prohibitive for real-world applications, requiring 2N+1 mutual information calculations for N features.
>
> **Response:**
> We would like to refer the Reviewer to Appendix G, where we discuss computational complexity in detail. While GIB requires $2N+1$ MI calculations at the input layer, this cost is independent of network depth, unlike standard IB which scales with the number of layers. Furthermore, GIB benefits from the ability to apply dimensionality reduction techniques, such as PCA to high-dimensional inputs, since features at the input layer share a common representation space. This is less straightforward for IB, which operates across layers with heterogeneous representations. We acknowledge that the input dimensionality (e.g., 3072 for CIFAR-10) can be high, but the one-time cost at the input layer compares favorably to the per-layer costs of IB when analyzing deep networks.
>
>
> ---
>
> ### Comment 2
> **Reviewer:**
> Mutual information estimation via binning is known to be inaccurate (Saxe et al., 2019), yet all major conclusions rely on this flawed estimation method.
>
> **Response:**
> We would like to thank the reviewer for the comment. We now draw major conclusions from both binning and KDE results. Please refer to Appendix F, where we show that for both KDE and GCMI estimation GIB more often leads to compression.
>
> ---
>
> ### Comment 3
> **Reviewer:**
> No comparison with other state-of-the-art generalization techniques (NIB [1], HSIC [2], DIB [3], Gate-IB [4]) or information-theoretic approaches.
>
> **Response:**
> We acknowledge that our original version of the manuscript did not sufficiently discuss alternative methodologies, and we have now revised the related work section to address this more thoroughly. We added an in-depth discussion of the papers listed by the Reviewer in that section.
>
> ---
>
> ### Comment 4
> **Reviewer:**
> Incorporating the concept of synergy lacks sufficient novelty...
>
> **Response:**
> We thank the reviewer for highlighting these papers and clarify that, while they are closely related, none of them reformulate the information bottleneck itself through the lens of synergy. Please find a discussion of each paper below.
>
> - **[1] Kolchinsky et al. Nonlinear Information Bottleneck**: This work uses the classical IB objective and does not introduce a synergy based IB.
> - **[2] Wang et al. HSIC Bottleneck**: This method replaces mutual information with HSIC but still optimizes a standard IB (without synergy).
> - **[3] Yu et al. Deep Deterministic Information Bottleneck (DIB)**: This paper uses matrix-based Rényi entropy to estimate MI under standard IB and does not separately define or optimize synergy.
> - **[4] Alesiani et al. Gated Information Bottleneck (Gate-IB)**: This work adds a gating mechanism to a conventional IB loss, no discussion of synergy.
> - **[5] Yu et al. Multivariate Extension of Matrix-Based Rényi's Entropy**: This paper does discuss synergy but not in the context of the IB.
> - **[6] Yu et al. Measuring Dependence with Matrix-Based Entropy Functional**: This work introduces multivariate dependence measures that measure synergy, but did not use this to update the IB.
> - **[7] Zhang et al. Multi-view Information Bottleneck**: This method applies matrix-based entropy to a supervised multi-view IB objective, where any synergy across views is implicit in the joint mutual information and not an explicit term in the objective or the discussion.
>
> Our work distinguishes itself from the above by explicitly formulating and optimizing a synergy-based term, rather than relying on synergy as an implicit characteristic of multivariate interactions. We would be very interested to see future work that re-designs the IB framework (particularly the compression term) with the explicit aim of investigating the role of synergy.
>
> ---
> ### Comment 5
> **Reviewer:**
> The paper demonstrates that GIB exhibits compression phases where standard IB fails, but does this translate to measurable improvements in test accuracy or other standard generalization metrics? Can you provide a quantitative analysis?
>
> **Response:**
> We would like to direct the Reviewer’s attention to Figure 7, which provides supporting evidence for this point.
> ---
>
> ### Comment 5
> **Reviewer:**
> The PCA preprocessing used for CIFAR-10 experiments potentially discards important synergistic relationships. Have you validated that the reduced representation preserves the synergistic interactions you aim to measure?
>
> **Response:**
> Appendix H shows that increasing the PCA dimensionality from 25 to 50 yields clearer and less noisy results. Beyond 50 dimensions, the improvements are minimal, suggesting that a PCA dimension of 50 is sufficient to preserve the relevant synergy.

---

### Official Review · Reviewer_TtzU · 2025-11-01

**Soundness:** 2
**Presentation:** 2
**Contribution:** 3
**Rating:** 2
**Confidence:** 3

**Summary:**

This work extends the traditional Information Bottleneck (IB) principle to account for synergistic information. This extension results in the introduction of the Generalized IB (GIB), which is claimed to be more capable of capturing compression phases in Deep Neural Networks (DNNs), while additionally being less prone to the problem of infinite MI values in deterministic DNNs. The authors conduct extensive experiments revealing compression phases under numerous setups, in which traditional IB fails to detect compression. The study supports these findings with both theoretical justification and informal explanations.

**Strengths:**

The experimental results are extensive, covering settings from toy problems to vision and language models. The resulting plots (at least for the binning estimator) clearly indicate a compression phase, even under conditions where IB compression does not typically occur (e.g., with ReLU activations). Overall, from my point of view, these findings are highly interesting.

**Weaknesses:**

**Methodological problems:**

The work is currently divided into two distinct parts: a theoretical component, which uses information theory to introduce the GIB and establish its properties (e.g., non-vacuousness and its connection to the original IB); and an experimental component, which relies on binning for mutual information (MI) estimation (or on a loss comparison in Appendix F).

For these parts to form a coherent whole, the MI estimates **must** be tight, or at least reflect the correct dynamics (i.e., the true MI increases if and only if the estimated value increases). However, based on established literature, I believe this connection is tenuous at best. This fundamental weakness calls the information-theoretic explanation of the observed phenomena into serious question, unless the authors can rigorously demonstrate the validity of their estimators.

Here are detailed explanations of my concerns:
1. **Binning estimator** is the most crude approach to MI estimation, which is notoriously problematic and misaligned with the actual MI in various setups.
    - This estimator is essentially the rectangle method applied to the differential entropy estimation (Theorem 8.3.1 in Cover & Thomas, 1991). Even under the infinite-samples assumption, this method is cursed with the dimensionality: given the integrated function is Lipschitz, the local approximation error can grow as $O(\sqrt{d})$; however, since we need $O(const^d)$ evaluations, the global error grows exponentially with $d$.
    - The problem worsens if we consider finite-sample regimes (Geiger, 2021):
       - If binning is too coarse and all samples fall into one bin, one has $\hat{H}(X) = \log (\text{bin size})$, while true $H(X)$ can vary from $-\infty$ to $\hat{H}(X)$.
       - If binning is too fine, one has $\hat{H}(X) = \log(\text{num. of samples}) - const$, while $H(X)$ can range from $-\infty$ to $\infty$. Moreover, in high dimensions, this is a very likely scenario for non-compressed data, as the probability to hit the same bin twice decays as $O(const^{-d})$.
       - Adaptive bin size selection has limited success in dealing with the issues above (Geiger, 2021).
    - Experimental evaluation confirms extremely poor performance of histogram-based MI estimators [1]
    - The resulting estimates **and** corresponding dynamics depend severely on the binning strategy. It has been shown changing the bin size or performing non-uniform binning can create or disappear the compression phase ([2] and Goldfeld, 2018; Saxe, 2018; Geiger, 2021).
2. **Loss Comparison**, while being more accurate, is only tight when the conditional probability is Gaussian (for MSE) or Laplace (for MAE); please, refer to Theorem 8.6.6 in (Cover & Thomas, 1991) for a general idea. This approximation is decent for low-dimensional cases. However, in higher dimensions, especially under the manifold hypothesis, the gap between estimated and real values of MI can be arbitrary high (since actual $H(Y \mid X)$ might $\to -\infty$ for any fixed MSE).
3. **Difference in the results**. I consider the plots in the main text and in the Appendix F to be quite different. Combined with the reasoning above, this makes me suspect that observed compression may arise from MI estimation artifacts in *both* cases.

[1] Czyz et al. "Beyond Normal: On the Evaluation of Mutual Information Estimators". Proc. of NeurIPS 2023.

[2] Noshad et al. "Scalable Mutual Information Estimation using Dependence Graphs". Proc. of ICASSP 2019.

**Theoretical Problems:**

Some steps in theoretical derivations are hard to follow.
1. Since $Q(Z,Y)$ is defined as a *distribution* (line 284), the notation $I(\mathcal{X};Q(Z,Y))$ is incorrect: MI is defined between random variables, not distributions. I kindly ask to clarify this part. If, however, $Q$ is instead defined as a PMI function applied to $Z$ and $Y$, I do not understand the notation $Q(Z,Y) = Z = Y$: in this case, $Q$ is 1-dimensional, while $Z$ is of arbitrary dimensionality.
2. While Theorem 3 seems fine to me (with an exception for $Q(Z,Y) = Z = Y$, which I am unable to follow at the moment), the converse result in lines 788--792 is rather informal and, probably, incorrect. There are many cases in Information Theory when $\infty - \infty \neq 0$.

    For instance, consider $I(X;Y;Z) = I(X;Y) - I(X;Y \mid Z)$ for $X = Y \sim U[-1;1]$ and $Z = {X > 0} \sim Bern(1/2)$. Here, both $I(X;Y)$ and $I(X;Y \mid Z)$ are infinite. But one can also write $I(X;Y;Z) = I(Z;Y) - I(Z;Y \mid X) = 1 \\; \text{bit} - 0 \\; \text{bit} = 1 \\; \text{bit}$.

    Moreover, by selecting $X = Y \sim \mathcal{N}(0, I_2)$ and $Z = X_1$, one gets $I(X;Y;Z) = I(X;Y) - I(X;Y \mid Z) = \infty - \infty$, but also $I(X;Y;Z) = I(Z;Y) - I(Z;Y \mid X) = \infty - 0 = \infty$.

**Minor issues:**
1. Time travel evidence in line 175: (Rosas et al., 20019).

**Conclusion:**

While I am both fascinated and intrigued by the experimental results, my concerns in regards with the methodology are strong and thus keep me from advocating for publishing this manuscript.

Currently, there appears a huge gap between the theory and practice. Additionally, there are some major inconsistencies in the experimental results and a certain level of informality in the theoretical derivations.

However, **I am more than willing to raise my score** above the acceptance treshold if the following happens during the rebuttal or revision: either the authors provide solid, rigorous and convincing arguments proving that the histogram-based estimator truly captures dynamics of MI, or the authors derive a separate (non-information-theory-based) theoretical explanation of the compression phenomenon from the perspective of the binning estimator.

**Questions:**

1. Please, clarify, what is $Q$, and how it is possible that $Q(Z,Y) = Z = Y$.
2. I kindly ask you to reinterpret the infinities in GIB given my remarks from the weaknesses section. Does one of the key claims (finiteness of GIB under reasonable settings) still hold?
3. Since there is no supplementary code attached, I also kindly ask to provide strict definitions/algorithms for both MI estimators. Please, also provide corresponding experimental protocols.

---

> ### Author Response · Authors · 2025-11-27
> **Response**
>
> ### Comment 1
> **Reviewer:**
> Binning estimator is the most crude approach to MI estimation, which is notoriously problematic and misaligned with the actual MI in various setups.
> * This estimator is essentially the rectangle method applied to the differential entropy estimation (Theorem 8.3.1 in Cover & Thomas, 1991). Even under the infinite-samples assumption, this method is cursed with the dimensionality: given the integrated function is Lipschitz, the local approximation error can grow; however, since we need evaluations, the global error grows exponentially.
> * The problem worsens if we consider finite-sample regimes (Geiger, 2021):
>    * If binning is too coarse and all samples fall into one bin, one has issues, while true MI can vary significantly.
>    * If binning is too fine, one has other issues, while MI can range widely. Moreover, in high dimensions, this is a very likely scenario for non-compressed data, as the probability to hit the same bin twice decays.
>    * Adaptive bin size selection has limited success in dealing with the issues above (Geiger, 2021).
> * Experimental evaluation confirms extremely poor performance of histogram-based MI estimators [1].
> * The resulting estimates and corresponding dynamics depend severely on the binning strategy. It has been shown changing the bin size or performing non-uniform binning can create or disappear the compression phase ([2] and Goldfeld, 2018; Saxe, 2018; Geiger, 2021).
>
> **Response:**
> We thank the reviewer for their insightful comments. Accordingly, we have revised the appendix to report results re-computed using both GCMI and KDE estimators. The main paper’s results generally align with the KDE outcomes: GIB tends to show compression more consistently.
>
> ---
>
> ### Comment 2
> **Reviewer:**
> Loss Comparison, while being more accurate, is only tight when the conditional probability is Gaussian (for MSE) or Laplace (for MAE); please refer to Theorem 8.6.6 in (Cover & Thomas, 1991) for a general idea. This approximation is decent for low-dimensional cases. However, in higher dimensions, especially under the manifold hypothesis, the gap between estimated and real values of MI can be arbitrarily high (since actual MI might vary for any fixed MSE).
>
> Difference in the results: I consider the plots in the main text and in the Appendix F to be quite different. Combined with the reasoning above, this makes me suspect that observed compression may arise from MI estimation artifacts in both cases.
>
> **Response:**
> We completely agree with the comments of the reviewer. Consequently, we have replaced these results with both GCMI and KDE MI estimation results.
>
> ---
>
> ### Comment 3
> **Reviewer:**
> Since Q is defined as a distribution (line 284), the notation is incorrect: MI is defined between random variables, not distributions. I kindly ask to clarify this part. If, however, Q is instead defined as a PMI function applied to $Z$ and $Y$, I do not understand the notation: in this case, one is 1-dimensional, while the other is of arbitrary dimensionality.
>
> **Response:**
> We agree with the reviewer: we have updated the paper to make it clear that the random variable $Q$ is sampled from the distribution $Q(Z;Y)$, and $Q$ has replaced $Q(Z,Y)$ in the text where appropriate. $Q = Y = Z$ if the model's outputs = $Y$.
>
> ---
>
> ### Comment 4
> **Reviewer:**
> While Theorem 3 seems fine to me (with an exception for one part, which I am unable to follow at the moment), the converse result in lines 788–792 is rather informal and, probably, incorrect. There are many cases in Information Theory when this does not hold.
>
> **Response:**
> We completely agree with the assessment of the Reviewer. Theorem 3 has been updated accordingly in the revised version of the paper.

---

### Official Review · Reviewer_py7F · 2025-11-01

**Soundness:** 2
**Presentation:** 3
**Contribution:** 3
**Rating:** 4
**Confidence:** 2

**Summary:**

In this work, the authors present the Generalized Information Bottleneck (GIB), a successor to the traditional Information Bottleneck (IB). Unlike IB, GIB is designed to overcome key limitations, such as producing vacuous infinite values in deterministic neural networks and lacking pronounced compression phases with non-tanh activations. Through extensive experiments, the authors demonstrate that GIB consistently reveals strong compression in DNNs.

**Strengths:**

GIB presents an original and appealing solution to IB's limitations. The authors strongly motivate its use through both theoretical analysis and comprehensive experiments. By connecting GIB to the established concept of synergy, they also provide an intuitive interpretation for the novel compression term they introduce.

**Weaknesses:**

1. In this work, the binning technique for mutual information (MI) estimation is utilized for the majority of the experiments. While the results appear consistent with the authors' intuition, the validity of these estimates is questionable. The histogram estimator is known to be among the least accurate non-parametric methods, even in low-dimensional settings [R1]. Furthermore, Goldfeld et al. (2019) have demonstrated that binning-based estimates can exhibit dynamics contrary to the true MI.
2. In Appendix F, the authors employ an alternative MI estimation method. From my perspective, the results differ significantly from the corresponding plots in the main text, which raises concerns about the robustness of the findings.
3. The definition and motivation for introducing $Q(Z,Y)$ are unclear to me. Its role in the formulation is not well-justified.
4. While I follow the proof of Theorem 3, the converse result (lines 787-793) is problematic. The derivation relies on an informal manipulation of the expression $\infty - \infty$, which is not mathematically rigorous to say the least.

[R1] Czyz et al., Beyond Normal: On the Evaluation of Mutual Information Estimators, NeurIPS 2023

**Questions:**

1. Can you, please, elaborate on the introduction of $Q$?
2. How do you explain the significant discrepancies between the MI dynamics obtained with the binning estimator and the loss-based estimator (shown in Appendix F)? Extending this observation, is it possible that using a more accurate MI estimator would reveal an absence of compression phases?
3. Can you provide a more formal derivation for the converse of Theorem 3?

---

> ### Author Response · Authors · 2025-11-27
> **Response**
>
> ---
>
> ### Comment 1
> **Reviewer:**
> In Appendix F, the authors employ an alternative MI estimation method. From my perspective, the results differ significantly from the corresponding plots in the main text, which raises concerns about the robustness of the findings.
>
> **Response:**
> We have now re-ran our results with both GCMI and KDE MI estimation techniques. In the case of KDE, we see that the results follow the same pattern of results as those obtained via binning in the main paper.
>
> ---
>
> ### Comment 2
> **Reviewer:**
> The definition and motivation for introducing Q(Z,Y) are unclear to me. Its role in the formulation is not well-justified.
>
> **Response:**
> $Q(Z,Y)$ is introduced so that we are able to capture synergy, while also ensuring the output in question is correct. Specifically, our complexity term is a function of a new distribution that describes the co-occurrences of $Z$ with $Y$. The exact definition of $Q(Z,Y)$ is based on PMI-based reweighting, i.e., weighting samples by the likelihood ratio between the joint distribution and the product of marginals: $Q(Z,Y) = P(Z,Y) / (P(Z)P(Y))$. This reweighting scheme emphasizes patterns where $Z$ and $Y$ co-occur more frequently than would be expected under independence, effectively highlighting the meaningful dependencies between our learned representations and the target outputs.
>
> ---
>
> ### Comment 3
> **Reviewer:**
> While I follow the proof of Theorem 3, the converse result (lines 787-793) is problematic. The derivation relies on an informal manipulation of the expression, which is not mathematically rigorous to say the least.
>
> **Response:**
> We fully agree with this concern and have updated the section accordingly; please refer to the revised PDF. We thank the reviewer for highlighting this.

---

### Official Review · Reviewer_nzFV · 2025-11-03

**Soundness:** 3
**Presentation:** 3
**Contribution:** 3
**Rating:** 6
**Confidence:** 3

**Summary:**

The paper proposes a Generalized Information Bottleneck (GIB) grounded in synergy—information that is only available through joint processing of features—to address well-known limitations of the classic Information Bottleneck (IB) in deep learning. Concretely, the authors (i) argue that synergistic functions generalize better than non-synergistic ones, with supporting theory and synthetic/empirical evidence; (ii) define a computable feature-wise synergy measure via average interaction information (II) and introduce a PMI-reweighted distribution \(Q(Z,Y)\) to emphasize correct predictions; and (iii) derive a Lagrangian objective (GIB) that promotes collective feature interactions while penalizing reliance on individual features. They further show that, under perfect estimation, the original IB objective is upper bounded by GIB, thereby ensuring compatibility with IB. Empirically, GIB exhibits robust compression phases across architectures and activations (including ReLU where standard IB often fails), yields interpretable training dynamics for CNNs/Transformers, and aligns with adversarial robustness trends.

**Strengths:**

**Originality.**
1. Recasts IB through the lens of feature synergy, operationalized via an II-based decomposition and a PMI reweighting that focuses the prediction term on correct outputs. This framing is distinct from prior multivariate IB/PID lines by directly optimizing for synergy rather than redundancy or total MI alone.

2.  Provides a principled rationale (theory $\to$ synthetic $\to$ CIFAR-10) that more synergistic functions have smaller MI with noise and inputs, yielding tighter generalization—connecting synergy to smoothness/Lipschitz arguments in known generalization bounds.

**Significance.**
Resolves practical roadblocks of IB (e.g., infinite/constant complexity in deterministic ReLU nets) and offers a unifying diagnostic that matches observed generalization/robustness behavior—potentially impactful for theory, interpretability, and robustness.

**Weaknesses:**

1. **Estimator dependence / computational load.** Although the paper justifies histogram binning for tractability and shows an alternative loss-based MI estimate (Appendix F), the synergy computation (feature-wise II with PMI reweighting) may be costly or sensitive to preprocessing (e.g., Kernel PCA to 50 dims on inputs for ResNets). More discussion on estimator bias/variance and runtime scaling would strengthen claims of practicality.


2. **Bounding assumptions.** The clean upper-bound result (IB $\le$ GIB) assumes perfect training accuracy/deterministic predictors; while standard in IB analyses, it would be valuable to extend the comparison to the realistic imperfect-estimation regime and quantify deviation.

**Questions:**

1. How sensitive are GIB’s information-plane conclusions to the MI estimator (binning vs. kNN/KDE/variational) and to dimensionality reduction choices (e.g., number/type of components in the ResNet experiments)? Can the authors report variability/error bars over estimators?

2. What is the time/memory cost of computing the feature-wise synergy decomposition per epoch on CIFAR-10/BERT, and how does it scale with feature dimension? Any approximations (e.g., random feature subsets) that preserve the qualitative planes?

---

> ### Author Response · Authors · 2025-11-27
> **Response**
>
> ---
>
> ### Comment 1
> **Reviewer:**
> Estimator dependence / computational load. Although the paper justifies histogram binning for tractability and shows an alternative loss-based MI estimate (Appendix F), the synergy computation (feature-wise II with PMI reweighting) may be costly or sensitive to preprocessing (e.g., Kernel PCA to 50 dims on inputs for ResNets). More discussion on estimator bias/variance and runtime scaling would strengthen claims of practicality.
>
> **Response:**
> We agree with this comment, and we have consequently updated the appendix to include GCMI and KDE results. We observe that the KDE results more closely correspond with those obtained through binning; specifically, we observe that GIB more often shows compression.
>
> ---
>
> ### Comment 2
> **Reviewer:**
> What is the time/memory cost of computing the feature-wise synergy decomposition per epoch on CIFAR-10/BERT, and how does it scale with feature dimension? Any approximations (e.g., random feature subsets) that preserve the qualitative planes?
>
> **Response:**
> Run times for both the PCA25 and PCA100 clusters turns out to be roughly the same, i.e., approximately 3 hours on an RTX 6000. As it appears, with the very quick binning method, training is the limiting factor.

---

### Author Response · Authors · 2025-11-27
**Overall Response on Paper Updates**

##  Summary of Changes: GIB Paper Revision

###  Key Change: Updated Prediction Term

The key methodological innovation is a **reformulation of the prediction term**, which more clearly separates the prediction and complexity components.

####  Previous Formulation

Both terms operated on the PMI-reweighted distribution $Q(Z,Y)$: $\mathcal{L}_{\text{GIB}} = I(\mathcal{X}; Q(Z,Y)) - \frac{1}{2\beta N} \sum_{i=1}^{N} \left(I(\mathcal{X}^{-i}; Q(Z,Y)) + I(X^i; Q(Z,Y))\right)$
####  New Formulation

The prediction term is now simply $I(Z; Y)$, the mutual information between model outputs and labels, while the complexity term remains unchanged: $\mathcal{L}_{\text{GIB}} = I(Z;Y) - \frac{1}{2\beta N} \sum_{i=1}^{N} \left(I(\mathcal{X}^{-i}; Q) + I(X^i; Q)\right)$

---

###  Why This Matters

This update **does not alter the core theory or interpretation of the GIB**. The fundamental insight remains unchanged: synergistic functions generalize better, and the GIB captures compression dynamics by measuring how features collectively (rather than individually) contribute to correct predictions. However, the change **makes the empirical effects more robust across MI estimators**. We now observe largely consistent effects between KDE and binning estimates.

---

###  New Appendix Evidence: KDE and GCMI Estimator Results

A major addition is **Appendix F** (renamed from "Alternative MI Estimation Using Loss Comparison" to "Alternative MI Estimation Using GCMI and KDE"), which now includes results using KDE and GCMI.

**Key finding:** With the updated prediction term, the **KDE estimator now exhibits similar dynamics to the binning estimator** on several experiments (activation functions, simple functions, BERT). This suggests there is an information-theoretic effect in the compression term, rather than an artifact of the binning procedure. The GCMI results are less interpretable.

---

###  Other Notable Changes

####  Expanded Related Work

The Related Work section on "Other Generalizations of the Information Bottleneck" has been expanded. The positioning relative to PID-based methods (Kolchinsky et al., 2022) has also been clarified: while prior work shows redundancy can be characterized via IB-style optimization, none explicitly incorporate a synergy-specific term into the IB formulation.

####  Updated Proofs

- **Theorem 1 proof** (Appendix B): Revised to reflect the new formulation, now starting from $I(Z;Y)$ rather than $I(\mathcal{X}; Q(Z,Y))$.
- **Sufficient statistics theorem** (Appendix C): Rewritten to show that maximizing $I(Z;Y)$ at $\beta=\infty$ recovers sufficient statistics when $H(Y|Z)=0$.
- **Infinity analysis** (Appendix D): Updated to reflect the new prediction term structure and avoid incorrect analysis.

####  Minor Text Refinements

- Section 3.2 now explicitly describes the two-step approach: (1) prediction term measures $I(Z;Y)$ directly, (2) complexity term uses $Q(Z,Y)$.
- We clarified that $Q$ is obtained by sampling from the PMI-reweighted distribution $Q(Z,Y) = \frac{P(Z,Y)}{P(Z)P(Y)}$ and replaced $Q(Z,Y)$ with $Q$ where appropriate.

---

###  Summary

The revision strengthens our empirical claims by showing that compression effects remain robust across different MI estimators (with binning and KDE exhibiting similar behaviour), while maintaining the integrity of the theoretical framework. Finally, updated text in the manuscript will be green.

---

### Author Response · Authors · 2025-12-04
**Follow Up Overall Response**

### Overall Response

First of all, we would like to thank the AC for their work and time. Since the rebuttal period was cut short, there was insufficient time for reviewers to engage with our responses and clarifications. One of the key criticisms we received was that our GIB does not afford robust interpretations across different MI estimation methods. However, in our responses we addressed this concern, showing that binning and KDE estimates give largely the same interpretation: synergy more readily shows compression. Please do not hesitate to contact us if you require further information.

The authors

---

### Note · Program_Chairs · 2026-01-17
**Submission Desk Rejected by Program Chairs**

The following references in this submission do not refer to real documents and/or have major errors in bibliographic information:

 Yanjun Han, Jiantao Jiao, and Tsachy Weissman. Differential entropy estimation via score
matching. In International Conference on Artificial Intelligence and Statistics (AISTAT’17),
2017.